# Interactive Learning for LLM Reasoning

## Abstract

Existing multi-agent learning approaches have developed interactive training environments to explicitly promote collaboration among multiple Large Language Models (LLMs), thereby constructing stronger multi-agent systems (MAS). However, during inference, they require re-executing the MAS to obtain final solutions, which diverges from human cognition that individuals can enhance their reasoning capabilities through interactions with others and resolve questions independently in the future. To investigate whether multi-agent interaction can enhance LLMs' independent problem-solving ability, we introduce ILR (**I**nteractive **L**earning for LLM **R**easoning), a novel co-learning framework for MAS that integrates two key components: Dynamic Interaction and Perception Calibration. Specifically, Dynamic Interaction first adaptively selects either cooperative or competitive strategies depending on question difficulty and model ability. LLMs then exchange information through Idea3 (Idea Sharing, Idea Analysis, and Idea Fusion), an innovative interaction paradigm designed to mimic human discussion, before deriving their respective final answers. In Perception Calibration, ILR employs Group Relative Policy Optimization (GRPO) to train LLMs while integrating one LLM's reward distribution characteristics into another's reward function, thereby enhancing the cohesion of multi-agent interactions. We validate ILR on three LLMs across two model families of varying scales, evaluating performance on five mathematical benchmarks and one coding benchmark. Experimental results show that ILR consistently outperforms single-agent learning, yielding an improvement of up to 5% over the strongest baseline. We further discover that Idea3 can enhance the robustness of stronger LLMs during multi-agent inference, and dynamic interaction types can boost multi-agent learning compared to pure cooperative or competitive strategies, providing useful insights toward future multi-agent design.

## 1 Introduction

Efforts to enhance the reasoning capabilities of Large Language Models (LLMs) have largely relied on training paradigms such as Supervised Fine-Tuning (SFT) (Achiam et al., 2023), Preference Learning (PL) (Rafailov et al., 2023), and Reinforcement Learning (RL) (Schulman et al., 2017; Guo et al., 2025; Zeng et al., 2025). These methods allow LLMs to iteratively interact with data and refine their behavior, essentially engaging in trial-and-error learning to acquire problem-solving skills, which can be viewed as self-learning for LLMs. However, real-world knowledge acquisition is rarely an isolated activity (Bloembergen et al., 2015; Canese et al., 2021). Humans continuously exchange knowledge through collaborative learning, as in peer discussions within classroom settings. While *single-agent learning* (self-learning) serves as the foundation of human education, *multi-agent learning* represents a more advanced and often more effective paradigm: multiple learners bring diverse perspectives, challenge each other's reasoning, and provide mutual feedback, ultimately leading to deeper understanding and more robust solutions (Kahveci & Imamoglu, 2007; Hsiung, 2012; Zambrano et al., 2019; Mende et al., 2021). The same principle suggests that multi-agent learning can benefit LLMs[1]: by exposing models to diverse reasoning strategies and peer-based feedback, it may help them overcome individual blind spots and develop stronger problem-solving abilities.

Recent studies have explored multi-agent learning. For example, MALT (Motwani et al., 2024) designs a sequential multi-agent system (MAS) consisting of Generator, Verifier, and Refiner agents, each independently trained to sample specialized trajectories. ReMA (Wan et al., 2025) introduces

---

[1]In our paper, "Agent" and "LLM" refer to the same entity/concept

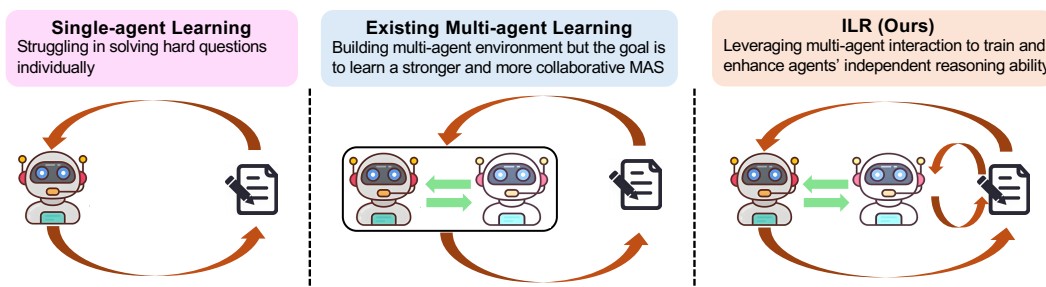

Figure 1: Conceptual comparison of ILR, existing Multi-agent Learning, and Single-agent Learning. ILR enhances LLMs' independent reasoning ability through Dynamic Interaction and Perception Calibration at training time, and LLMs can resolve questions independently at inference time.

a hierarchical framework with a high-level agent responsible for problem decomposition and a low-level agent for concrete step implementation, trained alternately to achieve complementary expertise. MAPoRL (Park et al., 2025) proposes a Post-Co-Training framework to enhance collaboration alignment through debate. However, during inference, these methods are required to re-execute the MAS to obtain final solutions, a process misaligned with human cognition, where individuals improve reasoning through peer interactions and subsequently solve problems independently.

In this paper, we address this gap by treating each agent as an autonomous entity and investigating whether multi-agent learning can enhance an LLM's individual problem-solving capacity (see Figure 1). We propose **ILR** (**I**nteractive **L**earning for LLM **R**easoning), a co-learning framework consisting of two key components: *Dynamic Interaction* and *Perception Calibration*.

The *Dynamic Interaction* module simulates human discussion. For "Dynamic", when confronted with complex problems, humans tend to cooperate, whereas for simpler problems, they often compete to identify the most efficient solution (Richard et al., 2002; Schneider et al., 2011). To emulate this behavior, an LLM estimates question difficulty through self-ranking and applies Item Response Theory (Cai et al., 2016; Benedetto et al., 2023) to calculate the probability of solving it independently. If the probability is low, the model engages in cooperation; otherwise, it chooses competition. For "Interaction", we design a novel **Idea3** framework, comprising three sequential stages: Idea Sharing (each LLM proposes its own solution), Idea Analysis (each LLM analyzes and reflects on the peer's solution), and Idea Fusion (the insights are synthesized into a refined and potentially novel solution). Following Dynamic Interaction, the *Perception Calibration* module is applied. Prior work (Ma et al., 2024; Park et al., 2025) has shown that incorporating tailored reward signals can effectively guide LLMs toward better multi-agent learning. Instead of relying on predefined hyper-parameters for reward shaping, we introduce a fully automated mechanism that integrates one LLM's reward distribution characteristics, derived from answer group sampling on the same input, into another LLM's reward function. We then employ Group Relative Policy Optimization (GRPO) (Shao et al., 2024) to update each LLM based on calibrated rewards. This plug-and-play calibration allows LLMs to perceive the quality of peer-generated solutions and adapt their reasoning accordingly.

We evaluate the effectiveness of ILR on two model families of different scales: Llama-3.1-8B-Instruct (Dubey et al., 2024), Qwen2.5-7B-Instruct, and Qwen2.5-14B-Instruct (Team, 2024). For multi-agent training, we pair these three models to form three different groups. Following (Zeng et al., 2025), we use MATH (Hendrycks et al., 2021) as the training dataset and assess performance across five mathematical reasoning benchmarks and a code benchmark. Experimental results demonstrate that ILR consistently outperforms traditional single-agent learning, achieving improvements of up to 5% over the strongest baseline models. Our investigation further reveals two findings: (1) Idea3 enhances the robustness of stronger LLMs during multi-agent inference scenarios. Analyzing and reflecting on the peer's solutions reduces the probability of being misled by weaker LLMs when exchanging information. (2) Dynamically determining interaction types can boost the efficacy of multi-agent learning and surpass pure cooperation or competition strategies. Our main contributions are summarized as follows:

- Unlike prior multi-agent learning approaches that primarily focus on improving inter-agent collaboration to strengthen overall system performance, to the best of our knowledge, we are the first to investigate whether multi-agent learning can more effectively enhance an LLM's independent reasoning capability compared to single-agent learning.

- Inspired by human interaction, we design a novel multi-agent learning framework ILR with Dynamic Interaction and Perception Calibration. The former adaptively selects cooperation or competition strategies and engages LLMs in Idea3 to gain better solutions. The latter enables LLMs to perceive the peer's performance and enhance the cohesion of multi-agent interactions.
- Through extensive experiments and analysis, we demonstrate the effectiveness of ILR over self-learning baselines. We further discover that Idea3 enhances the robustness of advanced LLMs by enabling them to analyze and reflect on the peer's solutions, and dynamic interaction strategies improve multi-agent learning by adapting to question difficulty.

## 2 RELATED WORK

Multi-agent learning (Busoniu et al., 2006; Han et al., 2024; Li et al., 2024b) first requires designing a multi-agent system that defines interaction paradigms among multiple agents, such as equi-level (Chan et al., 2023), hierarchical (Gronauer & Diepold, 2022), or nested structures (Zhao et al., 2025). Then, within this architectural framework, distinct agents engage in interactive sampling to acquire experience, which subsequently undergoes optimization through learning algorithms. Therefore, we systematically review prior works from the following two perspectives: Multi-Agent Communication (interactive paradigms) and Multi-Agent Training (optimization methods).

### 2.1 MULTI-AGENT COMMUNICATION

Traditionally, researchers employ recurrent neural networks (RNNs) as agents and utilize attention mechanisms to facilitate communication (Yu et al., 2019; Ding et al., 2024; Sun et al., 2024). For instance, TarMAC leverages multi-head attention to enable agents to learn both message content and targeted recipient (Das et al., 2019). After the emergence of LLMs, researchers develop numerous explainable prompt-based multi-agent communication. Notable examples include Debate, where multiple agents articulate arguments culminating in a final answer through majority voting mechanisms (Liang et al., 2023), and Actor-Critic, where actor agents generate solutions subsequently evaluated by critic agents through iterative feedback processes (Shinn et al., 2023; Estornell et al., 2024; Yuan & Xie, 2025). SiriuS introduces role specialization, assigning agents different professional identities (e.g., physicists, mathematicians) to sequentially solve problems while maintaining correct responses for fine-tuning and augmenting incorrect ones via feedback, regeneration, and rephrasing (Zhao et al., 2025). However, existing communication paradigms collectively conceptualize individual agents as components with optimization objectives centered on MAS performance.

In contrast, our work conceptualizes each agent as an independent entity and examines whether multi-agent interactions can enhance individual reasoning abilities. We emulate human discussion dynamics through a novel Idea3 interaction, specifically designed to facilitate critical thinking communication among agents via its three-stage process: Idea Sharing, Idea Analysis, and Idea Fusion.

### 2.2 MULTI-AGENT TRAINING

Conventional multi-agent training typically trains agents independently without awareness of other agents' states. Researchers employ multi-agent inference to collectively sample experiences, subsequently applying SFT or DPO to update individual agents independently, e.g., MALT (Motwani et al., 2024), Multiagent-FT (Subramaniam et al., 2025), and DEBATUNE (Li et al., 2024a). However, this static one-time sampling fundamentally compromises the dynamic nature of multi-agent interactions, wherein agents should engage in real time. To address this limitation, recent advances in Multi-Agent Reinforcement Learning (MARL) have enabled continuous, real-time interaction sampling among agents (Ma et al., 2024; Chen et al., 2025; Liao et al., 2025). For example, Liu et al. (2025) introduce Multi-Agent GRPO for training LLMs in multi-turn conversational settings. ReMA (Wan et al., 2025) proposes a turn-level ratio mechanism to alternately train high- and low-tier agents with multi-turn GRPO. MAPoRL (Park et al., 2025) implements a multi-agent proximal policy optimization algorithm, defining the agent state as the concatenation of interaction histories and incorporating manually predefined hyperparameters into rewards to incentivize collaboration.

Building on their design, we extend GRPO by introducing a fully automated reward calibration, where we incorporate distributional characteristics of rewards derived from agents' responses to the same question, enabling automatic peer perception without the need for manual intervention.

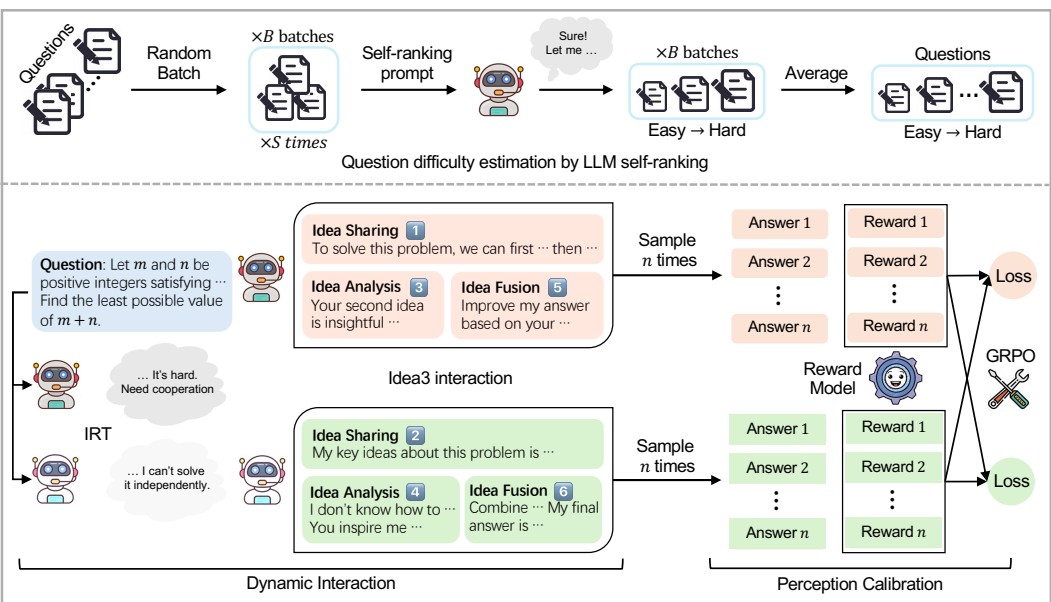

Figure 2: Illustration of proposed ILR for multi-agent learning. **Top**: LLMs first sort questions within each batch using the self-ranking prompt and then compute the average rank. **Bottom**: We depict the cooperation interaction here. If the question is too hard, LLMs will share their ideas, identify those complementary steps from other LLMs' ideas, and improve their answers. The competition interaction also follows Idea3, which only requires a minor change in prompts.

## 3 METHODOLOGY

As illustrated in Figure 2, our framework begins with each LLM receiving a question as input and estimating its difficulty using both self-ranking and Item Response Theory (IRT). Based on this estimation, LLMs dynamically select an appropriate interaction mode. They then engage in our proposed Idea3 interactions (Idea Sharing, Idea Analysis, and Idea Fusion), wherein distinct prompts guide different interaction modes, facilitating information exchange to generate the final answer. For each question, every LLM produces a group of answers. A reward model then evaluates these answers, after which we apply arithmetic calibration to encode the distributional characteristics of one LLM's rewards as incentive signals within another LLM's reward function. This mechanism allows LLMs to perceive the quality of their peers' solutions, fostering more effective multi-agent learning. Finally, we employ GRPO to optimize all participating LLMs under this learning paradigm. By doing so, we aim to more accurately simulate and study the human-like high-level behaviors in existing LLMs within a context that mirrors real-world human learning scenarios.

### 3.1 QUESTION DIFFICULTY ESTIMATION

In real-world learning environments, such as classrooms, students often adapt their strategies to the complexity of the problem: for challenging tasks, they are more likely to collaborate, whereas for simpler ones, they tend to compete to demonstrate the efficiency and superiority of their solutions (Richard et al., 2002; Green & Rechis, 2006; Schneider et al., 2011; Fülöp, 2022). Both cooperation and competition can serve as drivers of multi-agent learning, which provides the underlying motivation for our Dynamic Interaction design. Since most problems lack explicit or continuous difficulty annotations, we adopt a self-ranking to estimate the question difficulty based on the powerful comparative and ranking capabilities of LLMs (Wang et al., 2025).

Given a training dataset of $N$ questions, we divide it into $B$ random batches to avoid the long-context lost problem (Liu et al., 2023), with each batch containing $N' = \frac{N}{B}$ questions ($N' \ll N$). Using a self-ranking prompt (detailed in Appendix A.1), the LLM is instructed to order the questions within each batch in ascending difficulty. The ranks are then normalized into difficulty scores, where the easiest question is assigned $\frac{1}{N'}$ (rank 1) and the hardest one is assigned 1.0 (rank $N'$). Since a single random split provides only relative difficulty within its batch, we perform $S$ independent splits and

average their results to obtain a more stable and robust difficulty estimate for the entire dataset. We set $N'$ and $S$ to 10 in our work. For a set of LLMs $M = \{\mathcal{M}_i | i = 0, 1, ..., m\}$ in a multi-agent learning scenario, the difficulty score $D_q$ of a question $q$ is computed as:

$$D_q = \frac{1}{m} \sum_{i=1}^{m} \mathcal{M}_i \left( \frac{1}{S} \sum_{j=1}^{S} \frac{r_{q,j}}{N'} \right) \tag{1}$$

where $m$ is the number of LLMs, $\mathcal{M}_i(\cdot)$ denotes the estimation given by the $i$-th LLM, and $r_{q,j}$ is the rank of question $q$ in the $j$-th random split.

## 3.2 Dynamic Interaction

When an LLM with reasoning ability $\gamma_i$, which can be measured by its performance on a validation set, receives a question of difficulty level $D_q$, it can quantify the probability ($P_{q,i}$) of correctly answering the question using Item Response Theory (IRT) (Benedetto et al., 2023):

$$P_{q,i} = \frac{1}{1 + e^{-1.7 \times (\gamma_i - D_q)}} \tag{2}$$

where the empirically derived coefficient 1.7 has been shown to yield reliable predictions across diverse conditions (Baker, 2001; De Ayala, 2013; Benedetto et al., 2023). IRT is a mature framework from educational measurement designed to simultaneously model the relationship between three key variables: the ability of an individual, the difficulty of a question, and the probability of the individual correctly solving the question. This is highly analogous to our scenario: we aim to dynamically select an interaction strategy based on the LLM's ability and the problem's difficulty. Furthermore, previous works have investigated the intersection of IRT and AI/LLM (Wang et al., 2023; Lalor et al., 2024; Zhuang et al., 2023), demonstrating its relevance and utility in our domain. Since $P_{q,i} = 0.5$ when $\gamma_i = D_q$, we adopt 0.5 as the decision boundary between different interaction modes. We average the probability of $m$ LLMs to derive the overall probability ($P_q$) of independently solving question $q$ and determine the interaction mode.

$$P_q = \frac{1}{m} \sum_{i=1}^{m} P_{q,i} \tag{3}$$

$$\text{Mode} = \begin{cases} \text{Cooperation} & \text{if } P_q < 0.5 \\ \text{Competition} & \text{if } P_q \geq 0.5 \end{cases} \tag{4}$$

To simulate human discussion, we design a novel and unified three-stage Idea3 interaction for multi-agent communication: Idea Sharing (each LLM proposes its own solution), Idea Analysis (each LLM analyzes and reflects on the peer's solution), and Idea Fusion (the insights are synthesized into a refined and potentially novel solution). For different modes, we only need to slightly modify the prompt to inject the corresponding signal (details in Appendix A.1). Unlike current debate/reflection frameworks, which directly encourage one LLM to take another's output as advice without further thinking, our Idea3 is designed to foster critical thought within the LLM communication.

**Idea Sharing.** Each LLM begins by presenting its problem-solving strategy, explaining the reasoning process and methods employed to address the given problem. For example, when solving a complex algebraic equation, one model might focus on factoring, while another may rely on graphical analysis. This stage produces the *initial answer*.

**Idea Analysis.** Subsequently, LLMs engage in a critical evaluation of each other's proposed methods. In the cooperation mode, they may identify complementary strengths from different approaches, such as combining graphical insights with algebraic manipulation to generate a more comprehensive solution. In the competition mode, however, they rigorously assess the merits and limitations of the shared strategies. For example, one LLM might argue that the factoring approach, while effective, overlooks potential solutions that could be derived from the quadratic formula, thereby revealing a potential improvement.

**Idea Fusion.** Finally, LLMs synthesize the insights gained during previous analyses to generate a refined answer. This phase may involve integrating the most effective elements of both approaches,

yielding a solution that not only accurately addresses the original problem but also leverages complementary techniques from each LLM. For example, the final resolution to the algebraic equation might incorporate both the graphical representation for visual clarity and the algebraic methods for precision, culminating in a solution that is both robust and comprehensible. This structured interaction not only enhances the quality of the final response but also fosters a dynamic learning environment among LLMs, driving continuous improvement in their problem-solving capabilities. The output of this stage is the *updated answer*.

Prior research has noted that inter-agent communication may introduce noise into the information-exchange process and compromise the quality of final outputs (Pan et al., 2025; Zhang et al., 2025). To address this, we adopt a label-based selection mechanism: the initial answer is retained only if it is correct and the updated answer is incorrect; in all other cases, the updated answer is chosen. Finally, the rollout experience only contains a higher-quality step-by-step solution, and the reward is calculated solely based on this "final answer". This design choice can ensure: (1) To mitigate potential prompt shift. Since the rollout only contains a higher-quality step-by-step solution without intermediate results, the LLM is trained on the same type of data that it is expected to produce during inference. Therefore, there is no misalignment between the training and inference prompts. (2) To ensure a fair comparison with single-agent learning baselines. Including the rich information from the interaction trajectory in the reward signal could introduce a confounding variable, making it difficult to determine whether performance improvements stem from ILR or simply from the additional information in the trajectory.

## 3.3 PERCEPTION CALIBRATION

As demonstrated in (Park et al., 2025), incorporating incentive signals into rewards can effectively enhance both the cohesion of multi-agent interactions and the efficacy of multi-agent learning. However, Park et al. (2025) rely on adding manually predefined hyperparameter signals to rewards, which are discrete and coarse-grained, and in turn limits the scalability of reward shaping. In contrast, we introduce a fully automated method that integrates the distributional characteristics of one LLM's reward data into another LLM's reward function. This allows each model to perceive the quality of its peers' answers and generates continuous, fine-grained incentive signals without human intervention.

For a given input question, $m$ LLMs each perform $n$ sampling rounds, producing $m$ groups of responses, with $n$ answers in each group. A reward model is first used to assign initial rewards $R$ to all responses. Each group is then summarized by its maximum ($R_{max}$), minimum ($R_{min}$), and average ($R_{avg}$) scores, collectively reflecting the model's overall answer quality for that question. These statistics are arithmetically normalized and injected into the reward shaping process of peer models, yielding the final reward $\bar{R}$. For example, the $k$-th final reward of LLM $i$ is computed as:

$$\bar{R}_{i,k} = R_{i,k} + \sum_{l \in M \setminus \{\mathcal{M}_i\}} \text{clip}\left(\frac{R_{i,k} - R_{l,avg}}{R_{l,max} - R_{l,min}}, -\frac{1}{m-1}, \frac{1}{m-1}\right) \tag{5}$$

where clip($\cdot$) is a stabilization operation to prevent extreme values in reward shaping. Using these calibrated rewards, we then apply standard GRPO (Shao et al., 2024) to train and optimize LLMs.

## 4 EXPERIMENTS

### 4.1 EXPERIMENTAL SETTING

**Test Models.** We conduct experiments on three representative LLMs spanning two series and two scales: Llama-3.1-8B-Instruct (Dubey et al., 2024), Qwen2.5-7B-Instruct, and Qwen2.5-14B-Instruct (Team, 2024). To balance training cost with coverage, we organize these models into three pairwise groups and apply ILR training within each:

- Group1 (different series, same scale): Llama-3.1-8B-Instruct and Qwen2.5-7B-Instruct
- Group2 (different series, different scale): Llama-3.1-8B-Instruct and Qwen2.5-14B-Instruct
- Group3 (same series, different scale): Qwen2.5-7B-Instruct and Qwen2.5-14B-Instruct

**Benchmarks.** For ILR *training*, we use the MATH dataset (Hendrycks et al., 2021) following (Zeng et al., 2025). The MATH-500 subset (Hendrycks et al., 2021) is reserved for testing. From the

Table 1: The quantification comparison (accuracy %) of ILR and other baselines. ILR consistently outperforms all baselines, including single- and multi-agent learning, across all models.

| | GSM8K | MATH-500 | Minerva Math | Olympiad Bench | AIME 24&25 | Avg |
|---|---|---|---|---|---|---|
| Llama-3.1-8B-Instruct | 82.38 | 48.60 | 23.31 | 13.30 | 2.33 | 33.99 |
| SFT | 85.17 | 49.68 | 23.90 | 16.68 | 3.00 | 35.69 |
| DPO | 86.06 | 50.12 | 24.63 | 17.36 | 2.67 | 36.17 |
| PPO | 86.17 | 52.44 | 26.03 | 19.26 | 4.00 | 37.58 |
| GRPO | 85.70 | 52.32 | 26.69 | 19.79 | 4.00 | 37.70 |
| Reinforce++ | 86.88 | 51.40 | 29.78 | 18.52 | 3.33 | 37.98 |
| DebateFT-Group1 | 84.91 | 49.60 | 25.14 | 17.93 | 2.67 | 36.05 |
| DebateFT-Group2 | 84.82 | 49.24 | 24.19 | 16.89 | 2.67 | 35.56 |
| ILR-Group1 | **88.40** | **54.56** | 30.37 | **22.07** | **9.00** | **40.88** |
| ILR-Group2 | 87.38 | 54.36 | **32.28** | 21.51 | 6.67 | 40.44 |
| Qwen2.5-7B-Instruct | 91.83 | 74.84 | 40.95 | 36.83 | 9.33 | 50.76 |
| SFT | 91.90 | 75.92 | 41.25 | 37.24 | 11.33 | 51.53 |
| DPO | 92.11 | 75.72 | 42.43 | 37.54 | 10.33 | 51.63 |
| PPO | 92.12 | 76.44 | 43.31 | 38.07 | 11.67 | 52.32 |
| GRPO | 92.48 | 75.88 | 41.91 | 38.19 | 13.00 | 52.29 |
| Reinforce++ | 92.57 | 76.56 | 42.13 | 38.22 | 12.67 | 52.43 |
| DebateFT-Group1 | 92.16 | 75.96 | 41.25 | 37.28 | 10.33 | 51.40 |
| DebateFT-Group3 | 92.17 | 76.44 | 41.11 | 37.36 | 12.33 | 51.88 |
| ILR-Group1 | **93.00** | 77.00 | 44.56 | **39.50** | 16.33 | 54.08 |
| ILR-Group3 | 92.88 | **77.80** | **44.73** | 39.17 | **17.00** | **54.31** |
| Qwen2.5-14B-Instruct | 94.83 | 80.08 | 46.03 | 40.21 | 12.33 | 54.70 |
| SFT | 94.98 | 80.20 | 46.32 | 40.86 | 13.00 | 55.07 |
| DPO | 95.09 | 80.84 | 47.28 | 40.59 | 14.33 | 55.63 |
| PPO | 95.06 | 80.88 | 48.45 | 41.10 | 16.67 | 56.43 |
| GRPO | 94.86 | 80.48 | 47.43 | 42.11 | 16.00 | 56.17 |
| Reinforce++ | 95.03 | 80.68 | 48.31 | 41.39 | 16.67 | 56.41 |
| DebateFT-Group2 | 94.85 | 80.56 | 46.62 | 41.15 | 13.67 | 55.37 |
| DebateFT-Group3 | 94.92 | 80.76 | 47.57 | 41.48 | 12.00 | 55.35 |
| ILR-Group2 | **95.50** | 81.56 | 49.85 | 43.11 | **20.00** | 58.00 |
| ILR-Group3 | 95.42 | **82.32** | **50.07** | **43.53** | 19.66 | **58.20** |

remaining data, we randomly select 1,000 samples as the validation set to estimate each LLM's reasoning capability $\gamma_i$ (see Section 3.2), while the other 11,000 samples are used for training. For ILR **evaluation**, we conduct a comprehensive assessment across multiple mathematical reasoning benchmarks, which encompass both standard benchmarks, including GSM8K (Cobbe et al., 2021), MATH-500 (Hendrycks et al., 2021), Minerva Math (Lewkowycz et al., 2022), and Olympiad Bench (He et al., 2024), and a competition-level benchmark AIME24&25 (MAA-Committees, 2025), consisting of AIME2024 and AIME 2025. To further assess generalization beyond math, we evaluate ILR and baselines on a code generation benchmark MBPP (Austin et al., 2021). Note that except for Table 1, which is based on 5 random seeds, all the other results are based on seed 0.

**Baselines.** We compare our ILR with five single-agent learning (self-learning) baselines, including SFT (Achiam et al., 2023), DPO (Rafailov et al., 2023), PPO (Schulman et al., 2017), GRPO (Shao et al., 2024; Guo et al., 2025), and Reinforce++ (Hu, 2025), and a multi-agent learning baseline, DebateFT (Subramaniam et al., 2025). The selected single-agent learning baselines represent widely adopted and empirically effective approaches, with the latter three reinforcement learning algorithms demonstrating exceptional performance in recent LLM training. For multi-agent learning, direct comparisons are challenging because existing methods typically train specialized LLMs with complementary roles for problem-solving. To enable fair evaluation, we introduce a minor modification to Multiagent-FT (Subramaniam et al., 2025), sampling answers through Debate and optimizing each LLM using the original training algorithm. Each LLM will solve questions independently at the inference stage. We rename this baseline as DebateFT for clarity. We utilize Llama-3-8b-rm-

Table 2: Out-of-domain evaluation of ILR, DPO, and GRPO on MBPP (Pass@1). G$i$ means Group$i$. Compared with representative baselines, ILR further improves the coding ability of models.

| Model | Base | DPO | GRPO | ILR-G1 | ILR-G2 | ILR-G3 |
|---|---|---|---|---|---|---|
| Llama-3.1-8B-Instruct | 54.00 | 56.40 | 56.80 | 57.40 | **57.60** | - |
| Qwen2.5-7B-Instruct | 64.80 | 65.20 | 65.20 | 65.60 | - | **66.20** |
| Qwen2.5-14B-Instruct | 71.40 | 71.60 | 71.80 | - | **72.40** | 71.60 |

Table 3: Ablation Study of ILR. We report the average accuracy (%) of five mathematical evaluation benchmarks. DI, PC represent Dynamic Interaction and Perception Calibration.

| | Llama-3.1-8B-Instruct | | Qwen2.5-7B-Instruct | | Qwen2.5-14B-Instruct | |
|---|---|---|---|---|---|---|
| | Group1 | Group2 | Group1 | Group3 | Group2 | Group3 |
| ILR | **41.51** | **41.10** | **54.44** | **54.59** | **58.95** | **59.30** |
| DI-only | 39.12 | 39.25 | 53.95 | 53.23 | 58.04 | 58.57 |
| PC-only | 40.14 | 38.41 | 54.04 | 53.91 | 57.66 | 58.07 |

mixture (Hu et al., 2024) as the reward model to rate sampled answers. Implementation details about hyperparameters, training, and evaluation settings are provided in Appendix A.2.

## 4.2 Results and Analysis

### 4.2.1 LLMs Perform Better Through ILR Learning

**Overall Comparison.** Table 1 presents the quantitative comparison between ILR and other baselines. At the level of individual benchmarks, ILR achieves the best performance in nearly all cases, with the sole exception of Minerva Math on Qwen2.5-7B-Instruct, where PPO slightly outperforms it. In terms of average performance, ILR yields significant improvement in LLM reasoning ability. For example, Llama-3.1-8B-Instruct trained with ILR improves by 3.12% over the strongest baseline GRPO (41.51% vs. 38.39%), while both Qwen2.5 models also show gains of around 1% compared to their best-performing baselines. These results demonstrate that ILR's multi-agent learning framework consistently enhances the independent problem-solving capability of individual LLMs. For out-of-domain evaluation, we further compare ILR against two representative baselines, DPO and GRPO, and report the Pass@1 metric on the MBPP code benchmark (Austin et al., 2021). As shown in Table 2, ILR consistently achieves the strongest performance across all models, aligning with our findings on mathematical reasoning benchmarks.

From Table 1, we highlight two further insights: (1) **ILR Promotes Complex Reasoning.** On the competition-level dataset AIME24&25 (comprising AIME24 and AIME25), ILR significantly improves LLMs' capability in solving complex problems. For example, Llama-3.1-8B-Instruct with ILR doubles the performance of the best baseline GRPO (10.00% vs. 5.00%), while Qwen2.5-14B-Instruct improves by 3.33% over its strongest baseline (23.33% vs. 20.00%). These results demonstrate that both weaker and stronger models can benefit from ILR's multi-agent learning framework, enabling them to independently tackle challenging reasoning tasks. (2) **Balanced Grouping Improves Learning.** Each LLM is trained with ILR under two different grouping settings (see Section 4.1). Across comparisons, models achieve stronger results when paired with peers of more similar initial reasoning ability. For example, Llama-3.1-8B-Instruct performs better in ILR-Group1 with Qwen2.5-7B-Instruct (41.51% vs. 41.10% in Group2), as the two models are closer in initial capability. We attribute this phenomenon to the fact that excessive initial performance disparities may lead to imbalanced interactions where the stronger LLM overwhelmingly dominates the entire process, thereby compromising the quality of Dynamic Interaction. While this pattern consistently emerges, the observed performance differences are modest. We leave more comprehensive empirical validation of this finding to future research.

**Ablation Study.** To further analyze contributions of each component, we conduct an ablation study and report the average accuracy in Table 3. Removing either component leads to consistent performance drops across all models, underscoring their joint contribution to the overall performance.

Table 4: Multi-agent inference results on MATH-500. 'Single' denotes the single-agent inference performance of the base models, and * indicates the stronger LLM within each group. For Debate and Idea3, the better-performing result is highlighted in bold. ILR- means we utilize LLMs after ILR-training as base models.

| Inference | Llama-3.1-8B-Instruct | | Qwen2.5-7B-Instruct | | Qwen2.5-14B-Instruct | |
| Paradigm | Group1 | Group2 | Group1* | Group3 | Group2* | Group3* |
|---|---|---|---|---|---|---|
| Single | 49.80 | 49.80 | 75.60 | 75.60 | 81.20 | 81.20 |
| Debate | **64.00** | **66.20** | 74.60 | **80.00** | 79.20 | 81.00 |
| Idea3 (Ours) | 63.40 | 62.00 | **75.60** | 77.80 | **79.80** | **82.00** |
| ILR-Single | 55.80 | 55.20 | 77.60 | 78.00 | 81.80 | 82.60 |
| ILR-Debate | **65.60** | **71.20** | 81.20 | **81.00** | 79.40 | 81.20 |
| ILR-Idea3 (Ours) | 65.00 | 67.80 | **76.00** | 78.80 | **80.40** | **82.40** |

### 4.2.2 Dynamic Interaction Enhances Robustness of Stronger LLMs

In Section 4.2.1, we demonstrate how multi-agent learning through ILR training strengthens the independent reasoning abilities of individual LLMs. Here, we evaluate the effectiveness of Idea3 communication during the inference stage. In pure inference scenarios where ground-truth labels are unavailable, we employ a summarization prompt to synthesize the initial and updated responses, thereby mitigating noise from multi-agent interactions. Specifically, for a given input question, two LLMs first engage in Idea3 communication, after which each model evaluates both its own initial answer and the updated answer to produce a final prediction. For comparison, we also include a Debate paradigm, which treats other agents' outputs as additional advice to inform final answer generation. Full prompt details are provided in Appendix A.1.

Table 4 shows the multi-agent inference result on MATH-500 using untrained LLMs. We consistently observe that within different groups, Debate is more beneficial for weaker LLMs, while our Idea3 enhances the robustness of stronger LLMs by making them less susceptible to low-quality responses generated from weaker LLMs during multi-agent communication. We attribute this phenomenon to two primary reasons: **First**, for weaker LLMs, Debate directly incorporates stronger models' answers as additional guidance, enabling them to refine their outputs, which often results in a more significant improvement. **Second**, for stronger LLMs, Debate similarly compels them to consider answers from weaker LLMs, which are usually lower in quality and can potentially degrade performance. In contrast, our Idea3 prompts stronger LLMs to critically evaluate and selectively integrate peer contributions, filtering out noise and thereby improving robustness. We further conduct multi-agent inference using ILR-trained LLMs as base models. Results show that ILR-trained LLMs achieve better performance than untrained LLMs in the same multi-agent inference setup, which indicates that ILR does not diminish an LLM's ability to collaborate effectively at deployment.

### 4.2.3 Effect of Interactive Type (Cooperation or Competition)

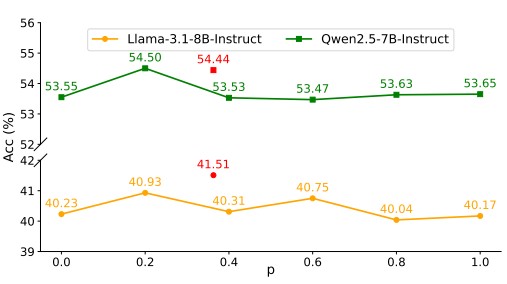

Figure 3: Average accuracy of Group1 under varying cooperation ratios. IRT is marked in red.

In ILR training, we employ Item Response Theory (IRT) to dynamically determine interaction types, i.e., cooperation or competition. To further investigate the influence of cooperation, we vary the cooperation ratio ($p$) from 0.0 to 1.0 in increments of 0.2. Here, $p = 0.0$ corresponds to full competition, $p = 1.0$ to full cooperation, and intermediate values designate the first $p$-proportion of questions (ranked by difficulty) as cooperative, with the remainder treated competitively. Due to training costs, we restrict this study to Group1.

Figure 3 shows the results, with IRT highlighted in red. Two key findings emerge for dynamic interaction design: (1) **Suboptimality of Extreme Strategies.** Relying solely on competition or co-

Table 5: Scaling trend of ILR (two vs. three LLMs per group). G$i$ denotes Group$i$. We report the average accuracy on five mathematical evaluation benchmarks.

| Model | Base | ILR-G1 | ILR-G2 | ILR-G3 | ILR-G4 |
|---|---|---|---|---|---|
| Llama-3.1-8B-Instruct | 34.15 | 41.51 | 41.10 | - | **42.48** |
| Qwen2.5-7B-Instruct | 51.76 | 54.44 | - | 54.59 | **55.12** |
| Qwen2.5-14B-Instruct | 55.57 | - | 58.95 | 59.30 | **59.95** |

operation is suboptimal for ILR, underscoring the necessity of adaptive interaction in multi-agent learning. This is intuitive: for challenging problems, cooperation allows LLMs to leverage complementary strengths and produce more comprehensive solutions, whereas for simpler tasks that can be effectively solved independently, excessive cooperation provides little benefit and may even introduce noise into the final outputs. (2) **Configuration of $p$.** The optimal cooperation ratio $p$ requires careful design. One option is to manually partition data into subsets and tune $p$, but this is costly. IRT offers a practical alternative by approximating problem difficulty and aligning it with model reasoning capability. Although not always optimal (e.g., for Qwen2.5-7B-Instruct), it achieves competitive results while eliminating manual intervention. This demonstrates the feasibility of IRT as a principled mechanism for integrating problem difficulty with LLM reasoning abilities. Future work may enhance robustness by incorporating additional conditional parameters into the IRT formulation.

### 4.2.4 SCALABILITY OF ILR

On balance of training costs and effectiveness, we primarily conduct experiments on three pairwise groups. To further evaluate the scalability of ILR beyond two-model interactions, we introduce a tri-model configuration, designated as Group 4 (ILR-G4). This experiment group integrates Llama-3.1-8B-Instruct, Qwen2.5-7B-Instruct, and Qwen2.5-14B-Instruct into a unified collaborative framework. The primary objective is to determine whether increasing the number of participating agents yields a positive scaling trend.

As evidenced in Table 5, the deployment of three LLMs results in a consistent and monotonic performance improvement across all constituent models compared to their dyadic counterparts (ILR-G1, G2, and G3). All LLMs surpass their best two-model performance by nearly 1.0%, which suggests that ILR effectively leverages the heterogeneity of larger model groups. The addition of a third model does not introduce noise or redundancy; rather, it provides complementary reasoning paths that further refine the consensus, confirming the robustness and positive scaling trend of our ILR. This presents an intriguing finding. Future work could focus on mitigating the training costs associated with multi-agent reinforcement learning, thereby enabling the investigation of interactive learning behaviors across broader ensembles of LLMs.

## 5 CONCLUSION

In this paper, we investigate whether interactive learning among multiple LLMs (*multi-agent learning*) can outperform traditional self-learning (*single-agent learning*). Unlike prior multi-agent learning approaches that assign complementary roles to agents within a MAS to maximize system-level performance, we treat each agent as an independent problem solver and aim to enhance individual capabilities. Inspired by real-world human interaction, we propose ILR, a novel framework built on two key components: Dynamic Interaction and Perception Calibration. Dynamic Interaction adaptively selects between cooperation and competition based on question difficulty and model ability, then enables information exchange among LLMs through the Idea3 paradigm (Idea Sharing, Idea Analysis, and Idea Fusion). Perception Calibration incorporates automated incentive signals into reward computation, allowing LLMs to perceive the quality of peers' answers and thereby strengthening multi-agent learning. Extensive experiments across different model series and scales demonstrate the effectiveness of ILR, showing that interactive learning consistently yields greater performance improvements than self-learning. We further discover that Idea3 improves the robustness of stronger LLMs, while dynamic interaction outperforms purely cooperative or competitive strategies. These findings align with human learning patterns and provide valuable insights into analyzing high-level, human-like behaviors in LLMs.

## 6 ETHICS STATEMENT

This study is conducted in strict accordance with the ethical guidelines outlined in the ICLR Code of Ethics and adheres to the principles of responsible AI research. Our research investigates whether multi-agent learning can enhance an LLM's individual problem-solving capacity through the proposed ILR (**I**nteractive **L**earning with **R**easoning) framework with Dynamic Interaction and Perception Calibration, which is inspired by human discussion patterns.

The research methodology is designed with ethical considerations, including rigorous bias mitigation protocols and fairness assessments throughout the experimental pipeline. For example, no content within our ILR is designed to target or disadvantage any demographic group, and we actively avoid discrimination by concentrating on both standard mathematical, competition-level mathematical, and code benchmarks that provide balanced coverage.

Importantly, this research does not involve any personal data or sensitive information. All training and evaluation benchmarks utilized in this study, such as GSM8K, MATH-500, Minerva Math, Olympiad Bench, AIME24&25, and MBPP, are publicly accessible, widely recognized within the AI community, and devoid of personally identifiable information. Data usage strictly adheres to the respective licenses governing these datasets. All research outputs, including code, data processing pipelines, and experimental results, are maintained with the highest standards of scientific integrity, transparency, and reproducibility in mind.

## 7 REPRODUCIBILITY STATEMENT

Our experiment is based on the open-source framework OpenRLHF (Hu et al., 2024). We attach our code, training data with a continuous difficulty level, and a README.md file with step-by-step instructions as supplementary materials to facilitate reproducibility. We also provide detailed information, such as self-ranking prompt, format prompt, Idea3 prompt, and evaluation prompt, in Appendix A.1. Additionally, we clarify some important hyperparameters, training, and evaluation settings like approximation time and GPUs in Appendix A.2.

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

## A    Experiment Details

### A.1    Prompt Details

---

**Self-ranking Prompt for Question Difficulty Estimation**

**Ranking Prompt1:**
Your task is to rank the given questions from easy to hard based on their difficulty level. Questions to be evaluated: $\{Q1, Q2, ..., QN'\}$.
**Ranking Prompt2:**
You will be given a batch of questions. Your task is to rank them from easy to hard based on their difficulty level. You should carefully horizontally compare the given questions in order to assign a suitable ranking place to each question. Please make sure you read and understand these instructions carefully. Please keep this document open while reviewing, and refer to it as needed. Questions to be evaluated: $\{Q1, Q2, ..., QN'\}$.
**Ranking Prompt3:**
You need to analyze and rank questions $Q1$-$QN'$ by their difficulty level based on these criteria: (1) Cognitive load required. If a problem involves multiple steps, it will have a higher cognitive load than a problem with a single step. (2) Knowledge depth needed. Problems involving the deduction of complex formulas require deeper professional knowledge. (3) Typical error rates. For example, problems that tend to overlook a certain prerequisite or calculation step will have a relatively high error rate. Please first score each question (1-10 scale) on three dimensions above, then calculate the average score, and rank these questions by final scores. Questions to be evaluated: $\{Q1, Q2, ..., QN'\}$.
**Format Prompt:**
After analyzing all the questions, please give all the ranking places (from easy to hard) in order, following the template "Ranking: [$Q$\{number of the easiest question\},..., $Q$\{number of the hardest question\} ]".

---

**Prompt for Idea3 communication**

**Idea Sharing (Cooperation&Competition):**
Question: {Input question}
Please reason step by step, and your final answer should be in the form boxed{answer} given at the end of your response.
**Idea Analysis (Cooperation):**
Partner's Contribution: {Ideas from other LLMs}
Collaboratively analyze the key steps in the partner's contribution, identify those steps that can help you improve your answer, and serve as additional advice.
**Idea Analysis (Competition):**
Opponent's Solution: {Ideas from other LLMs}
Critically analyze the opponent's ideas, identify the weaknesses and strengths of his ideas.
**Idea Fusion (Cooperation&Competition):**
Based on the above analysis, give an updated answer to the Original Question: {Input question}. Please reason step by step, and your final answer should be in the form boxed{answer} given at the end of your response.

---

**Prompt for Evaluation**

**Single-agent Evaluation:**
Please reason step by step, and your final answer should be in the form boxed{answer} given at the end of your response.
**Multi-agent Evaluation:**
**Debate:**
Here are solutions from other agents:
One agent response: {other_agent_response}
Using each response as additional advice based on the correctness of each response. Can you give an updated bullet-by-bullet answer to {Input question}. Please reason step by step, and your final answer should be in the form boxed{answer} given at the end of your response.
**Summarization (After Idea3 communication):**
The original question is {Input question}. There are two solutions you provided:
Solution 1: {Initial answer}
Solution 2: {Updated answer}
Please answer the original question step-by-step based on these two solutions, and your final answer should be in the form boxed{answer} given at the end of your response.

To mitigate potential prompt bias, we utilize three different self-ranking prompts of varying levels of granularity for each question and average the rankings to obtain the final estimation.

## A.2 IMPLEMENTATION DETAILS

**Training:** We use full-tuning to optimize the LLMs for one epoch. We use a batch size of 256 and a learning rate of 1e-6 for Llama-3.1-8B-Instruct, 1e-6 for Qwen2.5-7B-Instruct, and 9e-7 for Qwen2.5-14B-Instruct. The temperature is 0.5 for all LLMs, and the KL coefficient is 0 for Llama-3.1-8B-Instruct, 5e-7 for Qwen2.5-7B-Instruct, and 0 for Qwen2.5-14B-Instruct. The maximum output token number of the sampled answer is 2K. As for other hyperparameters, we strictly use the original parameters of GRPO. We utilize Llama-3-8b-rm-mixture (Hu et al., 2024) as the reward model to rate sampled answers. The approximate training time is 8 hours for Group1, 12 hours for Group2, and 10 hours for Group3. **Evaluation:** We set the temperature as 0 and the maximum output token number of evaluation is set to 8K for AIME, while 2K for other benchmarks. All training experiments are conducted on eight H100 GPUs, and evaluation experiments are conducted on one H100 GPU.

## A.3 STATISTICS OF BENCHMARKS

**Training Dataset**

Following Zeng et al. (2025), we only use **MATH** (Hendrycks et al., 2021) as our training data source for simplicity. Excluding the common MATH-500 (Hendrycks et al., 2021) as the evaluation set, there are 12000 samples in the remaining MATH dataset. We randomly select 1000 samples as the validation set to assess the LLMs' reasoning ability for Dynamic Interaction, while the remaining 11000 samples serve as the training set to fully fine-tune LLMs.

Each question of the MATH training set will have a continuous difficulty measured by LLMs' self-ranking, and we depict the question difficulty distribution in Figure 4. As illustrated in Section 4.1, the initial reasoning ability $\gamma_i$, which can be measured on the validation set, is 0.59, 0.75, and 0.78 for Llama-3.1-8B-Instruct, Qwen2.5-7B-Instruct, and Qwen2.5-14B-Instruct. According to the IRT in Section 3.2, the proportion of cooperation/competition across the three groups is as follows: 36.30/63.70%, 32.76/67.24%, and 16.87/83.13% for Group 1, Group 2, and Group 3, respectively.

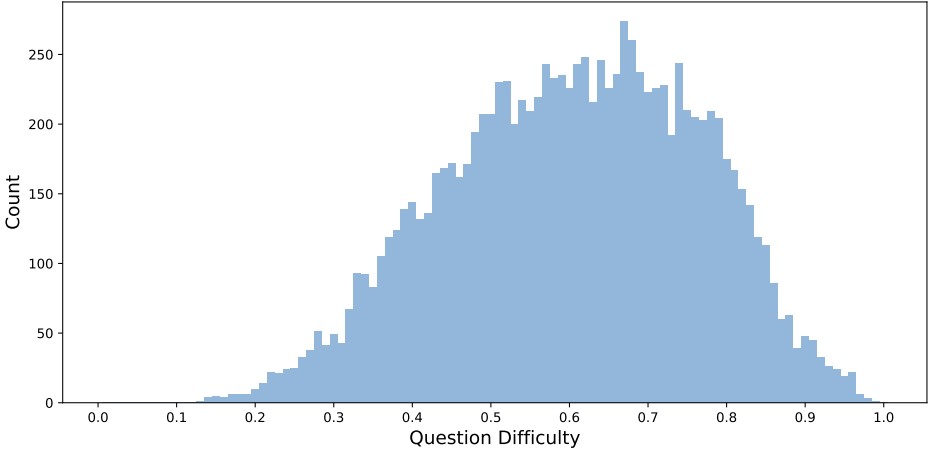

Figure 4: Question Difficulty Distribution (the interval is 0.01) of MATH training set used in ILR.

**Evaluation Dataset**

- **GSM8K** (Cobbe et al., 2021): This dataset comprises 1319 single-step mathematical reasoning questions designed for elementary school students in English. As one of the most widely adopted benchmarks in the field, it plays a basic role in assessing the arithmetic reasoning capabilities of LLMs.

- **MATH-500** (Hendrycks et al., 2021): The dataset encompasses 500 intermediate-level mathematical problems systematically curated across core domains, including algebra, geometry, precalculus, probability, and number theory. This carefully selected collection serves as a common evaluation framework to assess the comprehensive mathematical reasoning proficiency of LLMs.

- **Minerva Math** (Lewkowycz et al., 2022): This dataset contains 272 mathematical problems across core domains, including algebra, geometry, precalculus, probability, and number theory.

- **Olympiad Bench** (He et al., 2024): This benchmark constitutes a bilingual multimodal evaluation framework comprising 8476 Olympiad-level problems curated from prestigious mathematics and physics competitions. We utilize the subset processed by (Yang et al., 2024) as our test set, which contains 675 English text-only questions.

- **AIME24&25** (MAA-Committees, 2025): This benchmark collection contains 60 questions and derives from the 2024 and 2025 editions of the American Invitational Mathematics Examination (AIME), comprising two distinct problem sets. Each set contains 30 rigorously vetted mathematical questions characterized by high cognitive demand. The primary evaluative focus lies in probing advanced mathematical reasoning competencies, with particular emphasis on multi-faceted problem-solving strategies that require integration of complex conceptual frameworks.

- **MBPP** (Austin et al., 2021): This benchmark comprises 974 crowd-sourced Python programming problems, meticulously curated to align with the competency level of entry-level programmers. These problems encompass foundational programming concepts, standard library implementations, and essential algorithmic paradigms, ensuring a comprehensive evaluation of introductory programming proficiency. Following (Ni et al., 2023), we evaluate ILR and baselines using the test subset, which contains 500 questions.

## A.4 CLIP RATIO

In Section 3.3, we utilize a clip($\cdot$) function to stabilize the ILR training. In Table 6, we report the proportion of rewards that are actually clipped. It shows that the percentage of instances where the normalized reward difference exceeded the boundary is quite small. Therefore, we believe that this clip will not overly suppress the informative signals.

Table 6: The clip ratio of each group.

| Model | ILR-Group1 | ILR-Group2 | ILR-Group3 |
|---|---|---|---|
| Llama-3.1-8B-Instruct | 5.04% | 7.15% | - |
| Qwen2.5-7B-Instruct | 2.19% | - | 1.64% |
| Qwen2.5-14B-Instruct | - | 3.59% | 2.34% |

## B LLM USAGE

In preparing this paper, we utilize the large language model (LLM) primarily to polish texts for linguistic refinement and readability enhancement. The LLM-assisted revisions are as follows:

- **Grammatical Correctness**: We use the LLM to identify and rectify potential grammatical mistakes in our paper, such as subject-verb agreement errors, article misuse (e.g., definite/indefinite article selection), and prepositional phrase inconsistencies.

- **Stylistic Improvement**: We employ the LLM to enhance our linguistic expressions, such as instructing the LLM to refine, merge, and improve several simple sentences into well-structured and clear sentences.

All LLM-generated revisions undergo manual verification by the author, and the LLM is strictly limited to surface-level linguistic optimization without influencing conceptual frameworks or research conclusions.

## C   PSEUDOCODE OF ILR

---

**Algorithm 1** Interactive Learning for LLM Reasoning (ILR)

---

**Require:** Training set $\mathcal{D} = \{q_1, \ldots, q_N\}$, Validation set $\mathcal{D}_{val}$, Set of LLMs $\mathcal{M} = \{M_1, \ldots, M_m\}$,
    Reward model $\mathcal{M}_{reward}$, Self-ranking splits $S$, Batch size $N'$, Sampling rounds $n$
**Ensure:** Optimized Models $\mathcal{M}^*$
    ***Phase 1: Ability and Difficulty Estimation***
 1: Estimate model ability $\gamma_i$ for each $M_i \in \mathcal{M}$ based on accuracy on $\mathcal{D}_{val}$
 2: **for** each question $q \in \mathcal{D}$ **do**
 3:     Divide $\mathcal{D}$ into random batches of size $N'$ across $S$ splits
 4:     Collect ranks $r_{q,j}$ from $M_i$ via self-ranking prompt
 5:     $D_q \leftarrow \frac{1}{m} \sum_{i=1}^m M_i (\frac{1}{S} \sum_{j=1}^S \frac{r_{q,j}}{N'})$                    ▷ Calculate Difficulty (Eq. 1)
 6: **end for**
    ***Stage 2: Training Loop***
 7: **for** each batch $\mathcal{B} \subset \mathcal{D}$ **do**                    ▷ Iterate through the entire dataset (1 epoch)
 8:     $\mathcal{T} \leftarrow \emptyset, \mathcal{R} \leftarrow \emptyset, \bar{\mathcal{R}} \leftarrow \emptyset$
 9:     **for** each question $q \in \mathcal{B}$ **do**
10:         **1. Dynamic Interaction Strategy**
11:         Calculate success prob. $P_{q,i} \leftarrow \frac{1}{1+e^{-1.7(\gamma_i - D_q)}}$ for all $M_i$                    ▷ Eq. 2
12:         $P_q \leftarrow \frac{1}{m} \sum_{i=1}^m P_{q,i}$                    ▷ Eq. 3
13:         **if** $P_q < 0.5$ **then** Mode $\leftarrow$ Cooperation
14:         **else** Mode $\leftarrow$ Competition
15:         **end if**
16:         **2. Idea3 Interaction** (Sample $n$ times)
17:         **for** $k = 1$ to $n$ **do**
18:             **for** $i = 1$ to $m$ **do**
19:                 $a_{init} \leftarrow M_i.\text{Generate}(q, \text{Prompt}_{\text{Sharing}})$
20:                 $a_{ana} \leftarrow M_i.\text{Generate}(q, a_{init}^{\text{peer}}, \text{Mode}, \text{Prompt}_{\text{Analysis}})$
21:                 $a_{fus} \leftarrow M_i.\text{Generate}(q, a_{ana}, \text{Mode}, \text{Prompt}_{\text{Fusion}})$
22:                 **if** IsCorrect($a_{init}$) $\wedge \neg$IsCorrect($a_{fus}$) **then**
23:                     $a_{final} \leftarrow a_{init}$
24:                 **else**
25:                     $a_{final} \leftarrow a_{fus}$
26:                 **end if**
27:                 Form trajectory $\tau_{i,k}$ using $(q, a_{final})$, $\mathcal{T} \leftarrow \mathcal{T} \cup \{\tau_{i,k}\}$         ▷ Append trajectory
28:                 Record $R_{i,k}$ using $\mathcal{M}_{reward}(\tau_{i,k})$, $\mathcal{R} \leftarrow \mathcal{R} \cup \{R_{i,k}\}$         ▷ Append raw reward
29:             **end for**
30:         **end for**
31:         **3. Perception Calibration**
32:         **for** $i = 1$ to $m$ **do**
33:             **for** $k = 1$ to $n$ **do**
34:                 $V_{calib} \leftarrow 0$
35:                 **for** $l \in \mathcal{M} \setminus \{M_i\}$ **do**
36:                     Get peer stats $R_{l,max}, R_{l,min}, R_{l,avg}$
37:                     $V_{calib} \leftarrow V_{calib} + \text{clip}(\frac{R_{i,k} - R_{l,avg}}{R_{l,max} - R_{l,min}}, -\frac{1}{m-1}, \frac{1}{m-1})$
38:                 **end for**
39:                 $\bar{R}_{i,k} \leftarrow R_{i,k} + V_{calib}$                    ▷ Eq. 5
40:                 $\bar{\mathcal{R}} \leftarrow \bar{\mathcal{R}} \cup \{\bar{R}_{i,k}\}$                    ▷ Append calibrated reward
41:             **end for**
42:         **end for**
43:     **end for**
44:     **4. Optimization**
45:     **for** $i = 1$ to $m$ **do**
46:         Update $M_i$ via GRPO using trajectories $\mathcal{T}$ and calibrated rewards $\bar{\mathcal{R}}$
47:     **end for**
48: **end for**

---

Table 7: Comparison with z-score normalization (z-) based on Group1. We report the average accuracy (%) on five mathematical evaluation benchmarks and the clip ratio (%).

| Model | Avg | z-Avg | ratio | z-ratio |
|---|---|---|---|---|
| Llama-3.1-8B-Instruct | **41.51** | 40.74 ($\downarrow$ 0.77) | 5.04 | 20.45 ($\uparrow$ 15.41) |
| Qwen2.5-7B-Instruct | **54.44** | 53.94 ($\downarrow$ 0.50) | 2.19 | 9.10 ($\uparrow$ 6.91) |

## D  DISCUSSION

**Heterogeneous Role Assignments in ILR.** Our ILR intentionally employs a homogeneous design where all agents are problem solvers. This choice is directly tied to our core research question: to investigate whether and how interaction within a multi-agent environment can enhance the independent problem-solving capabilities of each individual LLM. In this context, all agents share the same role and learn from each other through interaction to achieve mutual improvement. However, we believe the idea of incorporating heterogeneous roles is interesting, and an exciting future direction would be to design a hybrid system. For example, one could have multiple "generator" agents that are trained internally via ILR to enhance their respective independent generation capabilities. Concurrently, the system could include one or more "verifier" or "refiner" agents responsible for evaluating and integrating the outputs from these ILR-strengthened generators. Such a hybrid model could potentially lead to an even more complex, robust, and powerful multi-agent system.

**Robustness of Dynamic Interaction.** For many common reasoning tasks, such as mathematics and standard question-answering, the intrinsic difficulty of a problem is one of the most direct and dominant factors influencing whether an individual chooses to persevere independently or seek collaborative assistance. However, in scenarios like creative writing, the "problem difficulty" does not necessarily correlate directly with the quality of the final output. Even a "simple problem" might benefit more from a cooperative strategy, as this could foster a richer blend of stylistic diversity by combining the unique perspectives of different LLMs. In such cases, the goal is not just to find a correct solution but to generate a high-quality, creative artifact. Therefore, for such specific task settings, we need to incorporate additional conditioning variables to improve the robustness of Dynamic Interaction.

## E  MORE EXPERIMENT RESULTS

### E.1  COMPARISON WITH Z-SCORE NORMALIZATION

To investigate the effect of different normalization methods in reward shaping(Section 3.3), we compare our min-max normalization with z-score normalization:

$$\bar{R}_{i,k} = R_{i,k} + \sum_{l \in M \setminus \{\mathcal{M}_i\}} \text{clip}(\frac{R_{i,k} - R_{l,avg}}{R_{std}}, -\frac{1}{m-1}, \frac{1}{m-1}) \tag{6}$$

We report the average accuracy on five mathematical evaluation benchmarks and the clip ratio discussed in Appendix A.4. Table 7 shows that implementing z-score normalization within the ILR framework yields inferior performance compared to our method. We attribute this degradation to the aggressive clipping associated with z-score normalization, which tends to suppress the magnitude of incentive signals. This observation further corroborates the superiority of our min-max normalization strategy.

### E.2  CONFIDENCE INTERVALS OF TABLE 1

In Table 8, we report the 95% confidence intervals of Table 1. Note that except for the result in Table 1, which is based on 5 random seeds, all the others are based on the result of seed 0.

Table 8: 95% Confidence Intervals of Table 1.

| | GSM8K | MATH-500 | Minerva Math | Olympiad Bench | AIME 24&25 | Avg |
|---|---|---|---|---|---|---|
| Llama-3.1-8B-Instruct | ±1.78 | ±1.33 | ±1.79 | ±0.83 | ±1.13 | ±0.69 |
| SFT | ±0.39 | ±1.61 | ±1.83 | ±1.12 | ±0.92 | ±0.68 |
| DPO | ±0.63 | ±1.48 | ±1.07 | ±0.94 | ±1.13 | ±0.18 |
| PPO | ±1.74 | ±1.62 | ±3.08 | ±0.93 | ±2.36 | ±0.93 |
| GRPO | ±0.72 | ±3.43 | ±1.82 | ±2.16 | ±2.36 | ±1.13 |
| Reinforce++ | ±1.16 | ±2.66 | ±1.65 | ±1.71 | ±2.27 | ±0.72 |
| DebateFT-Group1 | ±1.78 | ±3.01 | ±1.73 | ±2.40 | ±1.85 | ±1.85 |
| DebateFT-Group2 | ±0.73 | ±2.72 | ±4.32 | ±3.35 | ±1.85 | ±2.03 |
| ILR-Group1 | ±0.70 | ±0.97 | ±0.24 | ±0.18 | ±1.14 | ±0.55 |
| ILR-Group2 | ±0.21 | ±1.14 | ±1.18 | ±0.71 | ±1.46 | ±0.77 |
| Qwen2.5-7B-Instruct | ±0.57 | ±1.35 | ±1.84 | ±0.62 | ±2.78 | ±1.20 |
| SFT | ±0.86 | ±2.10 | ±1.63 | ±1.44 | ±3.70 | ±1.43 |
| DPO | ±0.62 | ±2.02 | ±1.50 | ±0.33 | ±3.40 | ±1.10 |
| PPO | ±0.83 | ±1.39 | ±2.03 | ±0.82 | ±2.07 | ±1.23 |
| GRPO | ±0.39 | ±1.12 | ±1.12 | ±1.05 | ±2.70 | ±0.97 |
| Reinforce++ | ±0.24 | ±0.88 | ±1.14 | ±0.53 | ±1.85 | ±0.80 |
| DebateFT-Group1 | ±0.54 | ±1.72 | ±2.88 | ±1.37 | ±0.93 | ±1.10 |
| DebateFT-Group3 | ±0.41 | ±0.59 | ±3.03 | ±1.08 | ±1.85 | ±0.86 |
| ILR-Group1 | ±0.31 | ±0.46 | ±1.18 | ±0.28 | ±2.26 | ±0.58 |
| ILR-Group3 | ±0.20 | ±0.25 | ±0.68 | ±0.28 | ±2.26 | ±0.48 |
| Qwen2.5-14B-Instruct | ±0.18 | ±1.45 | ±2.32 | ±1.18 | ±3.76 | ±1.13 |
| SFT | ±0.28 | ±1.02 | ±2.11 | ±1.17 | ±3.07 | ±1.23 |
| DPO | ±0.27 | ±0.79 | ±2.54 | ±2.39 | ±1.14 | ±0.95 |
| PPO | ±0.49 | ±1.33 | ±1.84 | ±1.91 | ±2.53 | ±1.43 |
| GRPO | ±0.72 | ±1.50 | ±2.28 | ±1.13 | ±3.14 | ±1.47 |
| Reinforce++ | ±0.25 | ±1.22 | ±0.95 | ±1.79 | ±4.86 | ±1.42 |
| DebateFT-Group2 | ±0.18 | ±0.97 | ±2.05 | ±1.11 | ±1.73 | ±0.30 |
| DebateFT-Group3 | ±0.56 | ±0.75 | ±2.63 | ±1.25 | ±3.70 | ±1.41 |
| ILR-Group2 | ±0.08 | ±0.21 | ±0.41 | ±0.43 | ±2.53 | ±0.72 |
| ILR-Group3 | ±0.14 | ±0.33 | ±0.99 | ±0.51 | ±2.70 | ±0.86 |

### E.3 COMPARISON WITH MORE SELF-LEARNING METHODS

In Table 1, we compare ILR with several basic self-learning methods like SFT and GRPO. Here, we discuss more self-learning baselines and compare ILR with them based on Llama-3.1-8B-Instruct (see Table 9).

**Self-reflection Learning.** Self-reflection and self-improvement constitute a pivotal research trajectory for enhancing agent capabilities through an intra-agent feedback loop (Renze & Guven, 2024; Shinn et al., 2023; Dou et al., 2024; Zhang et al., 2024). The central premise is that an agent can refine its performance by critically analyzing its own generated trajectories, answers, or reasoning processes in a closed-loop manner. To rigorously benchmark our multi-agent approach against this single-agent paradigm, we employ Re-ReST as a representative baseline. This comparison aims to determine whether the diverse feedback from peer agents offers superior gradients for improvement compared to the model's internalized self-verification.

**Advanced GRPO Variants.** Recent advancements in reinforcement learning have introduced sophisticated variants of GRPO (Lin et al., 2025; Sun et al., 2025), which significantly outperform the vanilla GRPO implementation. By comparing against CPPO (Lin et al., 2025), we seek to unveil whether the multi-agent enhancement provides a distinct advantage over purely algorithmic improvements in single-agent training stability.

Table 9: Comparison with Re-ReST, CPPO, and 2-GRPO, on Llama-3.1-8B-Instruct.

| | GSM8K | MATH-500 | Minerva Math | Olympiad Bench | AIME 24&25 | Avg |
|---|---|---|---|---|---|---|
| Llama-3.1-8B-Instruct | 82.87 | 49.80 | 22.79 | 13.63 | 1.67 | 34.15 |
| Re-ReST | 84.23 | 49.80 | 29.04 | 15.56 | 5.00 | 36.73 |
| CPPO | 84.84 | 54.60 | 27.57 | 18.85 | 6.67 | 38.51 |
| 2-GRPO | 86.43 | 53.20 | 28.31 | 18.22 | 6.67 | 38.57 |
| ILR-Group1 | **89.39** | **55.80** | 30.15 | 22.22 | **10.00** | **41.51** |
| ILR-Group2 | 87.26 | 55.20 | **33.82** | **22.52** | 6.67 | 41.10 |

Table 10: Comparison with periodic updates of LLM's reasoning ability $\gamma_i$ (periodic-) based on Group1. We report the average accuracy on five mathematical evaluation benchmarks and the proportion of cooperation/competition.

| Model | Avg | periodic-Avg | proportion | periodic-proportion |
|---|---|---|---|---|
| Llama-3.1-8B-Instruct | 41.51 | **42.35** ($\uparrow$ 0.84) | 36.30/63.70% | 32.44/67.56% |
| Qwen2.5-7B-Instruct | 54.44 | **54.82** ($\uparrow$ 0.38) | | |

**GRPO with More Sampling (Compute-Matched Baseline).** To rule out the possibility that performance gains are simply artifacts of increased sampling, we introduce a computation-aligned baseline: single-agent GRPO with doubled rollout numbers (denoted as 2-GRPO). Since our multi-agent setting involves interactions that effectively increase the inference load, this variant establishes a fair comparison by equating the sampling volume. By contrasting ILR with this high-sampling baseline, we aim to demonstrate that the benefits of our method stem from the quality of interactive feedback rather than from more sampling.

### E.4    COMPARISON WITH PERIODIC UPDATES OF LLM'S REASONING ABILITY

As a model's reasoning ability can indeed evolve during training. To investigate this, we conduct an additional experiment to study the impact of dynamically updating $\gamma_i$. Specifically, we re-evaluated $\gamma_i$ periodically every 5 training steps (we have a total of 42 steps). We report the average accuracy on five mathematical evaluation benchmarks and the proportion of cooperation/competition. Table 10 indicates that the performance of periodic-updates surpasses that of one-time measurement. As the reasoning capability gap diminishes, the cooperation rate declines, a dynamic that appears more organic, thereby further enhancing performance.

### E.5    COMPARISON WITH MIXED TYPES INTERACTION

In real human collaboration, cooperation and competition often coexist within the same reasoning process. To investigate this scenario, we modified our sampling procedure as follows: for a given problem, even when our IRT-based mechanism determines to use a cooperation strategy, we enforce a mixture of strategies across the multiple samples. For example, if we generated five distinct rollouts for a single question, we would randomly assign the designated cooperation strategy to 3-5 of these rollouts, while assigning the competition strategy to the remaining rollouts.

As shown in Table 11, this shift yielded asymmetric outcomes. Llama-3.1-8B-Instruct achieved a performance gain of 0.40%, likely because the mixed strategy exposed it to more cooperative reasoning paths on challenging problems. In contrast, Qwen2.5-7B-Instruct experienced a regression of 0.47%. This decline suggests that the "mixed" samples acted as noise for the stronger model, where forcing it to collaborate (when it should compete) likely constrained its reasoning potential or introduced errors from the weaker partner.

Table 11: Comparison with mixed types interaction (mixed-) based on Group1. We report the average accuracy on five mathematical evaluation benchmarks and the proportion of cooperation/-competition.

| Model | Avg | mixed-Avg | proportion | mixed-proportion |
|---|---|---|---|---|
| Llama-3.1-8B-Instruct | 41.51 | **41.91** (↑ 0.40) | 36.30/63.70% | 40.99/59.01% |
| Qwen2.5-7B-Instruct | **54.44** | 53.97 (↓ 0.47) | | |

Table 12: Quantitative analysis of computational overhead versus performance gains. We report the average accuracy on five mathematical evaluation benchmarks (%, without []) and the training time (hours, within []).

| Model | GRPO | ILR-Group1 | ILR- Group2 | ILR-Group3 |
|---|---|---|---|---|
| Llama-3.1-8B-Instruct | 38.39[3.75] | 41.51[8.00] | 41.10[12.00] | - |
| Qwen2.5-7B-Instruct | 53.58[3.25] | 54.44[8.00] | - | 54.59[10.00] |
| Qwen2.5-14B-Instruct | 57.55[5.50] | - | 58.95[12.00] | 59.30[10.00] |

Table 13: Quantitative comparison of the same LLM with different prompts versus different LLMs with the same prompt based on Llama-3.1-8B-Instruct.

| | GSM8K | MATH-500 | Minerva Math | Olympiad Bench | AIME 24&25 | Avg |
|---|---|---|---|---|---|---|
| Llama-3.1-8B-Instruct | 82.87 | 49.80 | 22.79 | 13.63 | 1.67 | 34.15 |
| Prompt1 | 87.79 | 54.20 | 29.41 | 20.30 | 6.67 | 39.67 |
| Prompt2 | 87.72 | 55.60 | 30.51 | 20.15 | 6.67 | 40.13 |
| ILR-Group1 | **89.39** | **55.80** | 30.15 | 22.22 | **10.00** | **41.51** |
| ILR-Group2 | 87.26 | 55.20 | **33.82** | **22.52** | 6.67 | 41.10 |

### E.6 COST TABLE

We compare the training time and performance of GRPO and ILR in Table 12. Note that the training durations reported for ILR-Group$i$ represent the aggregated time required for both LLMs within the group. For instance, in the ILR-Group1 setting, the combined training time for Llama-3.1-8B-Instruct and Qwen2.5-7B-Instruct amounts to 8 hours. In comparison, training these models individually using standard GRPO requires a total of 7 hours (3.75 hours for Llama-3.1-8B-Instruct and 3.25 hours for Qwen2.5-7B-Instruct). Consequently, incorporating ILR introduces only a marginal computational overhead of one hour. This slight increase in cost is justified by the performance gains, yielding improvements of approximately 3% for Llama-3.1-8B-Instruct and 1% for Qwen2.5-7B-Instruct.

### E.7 SAME LLM WITH DIFFERENT INPUT PROMPTS

In our primary experiments, we conceptualized distinct LLMs receiving the same input prompt as individual agents to investigate interactive learning among heterogeneous models. In this section, we extend this inquiry by examining the efficacy of interactive learning within a homogeneous setting, where a single LLM utilizes different input prompts to simulate distinct agents.

- **Prompt1**: $< Question >$ Please reason step by step.

- **Prompt2**: $< Question >$ Please double-check the question, first extract the keywords and think about the knowledge about these keywords, then solve the question.

Two notable findings emerge from this analysis (see Table 13). **First**, while ILR with an identical LLM still yields performance improvements over single-agent baselines, it does not surpass the performance of ILR with different LLMs. This suggests that while diversity can be simulated through varied prompting, distinct models provide a richer and more effective source of alternative perspectives. The interaction between truly heterogeneous agents appears to generate a more powerful learning signal. **Second**, the post-training performance of the Llama model utilizing Prompt2 was superior to its counterpart. This observation underscores that the granularity of instruction prompts plays a significant role in influencing the outcomes of interactive learning.

## F    CASE STUDY

We show some cases of **Dynamic Interaction (Idea3)** based on Group1 (LLM1: Llama-3.1-8B-Instruct and LLM2: Qwen2.5-7B-Instruct).

---

**Cooperation mode (both LLMs initially fail but succeed after interaction)**

**Problem:**
The function $f(x)$ satisfies $f(x) + f\left(\frac{x+1}{1-3x}\right) = x$ for all $x \neq \frac{1}{3}$. Find $f(2)$.

**LLM1 (Idea Sharing):**
To find $f(2)$, we can use the given functional equation and plug in $x = 2$. This gives us

$$f(2) + f\left(\frac{2+1}{1-3\cdot 2}\right) = 2 \Rightarrow f(2) + f\left(-\frac{3}{5}\right) = 2$$

Now, we can plug in $x = -\frac{3}{5}$ into the original functional equation to get

$$f\left(-\frac{3}{5}\right) + f\left(\frac{-\frac{3}{5}+1}{1-3\cdot\left(-\frac{3}{5}\right)}\right) = -\frac{3}{5} \Rightarrow f\left(-\frac{3}{5}\right) + f\left(\frac{1}{7}\right) = -\frac{3}{5}$$

Now, we can plug in $x = \frac{1}{7}$ into the original functional equation to get

$$f\left(\frac{1}{7}\right) + f\left(\frac{\frac{1}{7}+1}{1-3\cdot\frac{1}{7}}\right) = \frac{1}{7} \Rightarrow f\left(\frac{1}{7}\right) + f(2) = \frac{1}{7}$$

Now we have two equations and two unknowns. We can solve for $f(2)$ by subtracting the second equation from the first equation:

$$f(2) + f\left(-\frac{3}{5}\right) - \left(f\left(\frac{1}{7}\right) + f(2)\right) = 2 - \frac{1}{7} \Rightarrow f\left(-\frac{3}{5}\right) - f\left(\frac{1}{7}\right) = \frac{13}{7}$$

Now we can plug in $f\left(-\frac{3}{5}\right) = 2 - f\left(\frac{1}{7}\right)$ into the equation $f\left(-\frac{3}{5}\right) + f\left(\frac{1}{7}\right) = -\frac{3}{5}$ to get

$$2 - f\left(\frac{1}{7}\right) - f\left(\frac{1}{7}\right) = -\frac{3}{5} \Rightarrow f\left(\frac{1}{7}\right) = \frac{13}{10}$$

Now we can plug in $f\left(\frac{1}{7}\right) = \frac{13}{10}$ into the equation $f\left(-\frac{3}{5}\right) - f\left(\frac{1}{7}\right) = \frac{13}{7}$ to get

$$f\left(-\frac{3}{5}\right) - \frac{13}{10} = \frac{13}{7} \Rightarrow f\left(-\frac{3}{5}\right) = \frac{221}{70}$$

Now we can plug in $f\left(-\frac{3}{5}\right) = \frac{221}{70}$ into the equation $f(2) + f\left(-\frac{3}{5}\right) = 2$ to get

$$f(2) + \frac{221}{70} = 2 \Rightarrow f(2) = -\frac{81}{70}$$

The final answer is: $\boxed{-\dfrac{81}{70}}$. ✗

**LLM2 (Idea Sharing):**
To solve for $f(2)$ given the functional equation

$$f(x) + f\left(\frac{x+1}{1-3x}\right) = x$$

---

We need to find a way to express $f(x)$ in a form that allows us to determine $f(2)$. First, let's denote the transformation $g(x) = \frac{x+1}{1-3x}$. We need to find the inverse of $g(x)$ to understand the behavior of the function under repeated applications. To find the inverse, we solve for $x$ in terms of $y$ where $y = g(x)$:

$$y = \frac{x+1}{1-3x} \Rightarrow x = \frac{y-1}{1+3y}$$

Thus, the inverse function is

$$g^{-1}(x) = \frac{x-1}{1+3x}$$

Now, we apply the functional equation twice. First, we substitute $x$ with $g(x)$:

$$f(g(x)) + f\left(\frac{g(x)+1}{1-3g(x)}\right) = g(x)$$

Since $g\left(\frac{x+1}{1-3x}\right) = x$, we have:

$$f(g(x)) + f(x) = g(x)$$

Next, we substitute $x$ with $g^{-1}(x)$:

$$f(g^{-1}(x)) + f\left(\frac{g^{-1}(x)+1}{1-3g^{-1}(x)}\right) = g^{-1}(x)$$

Since $g\left(\frac{x-1}{1+3x}\right) = x$, we have:

$$f(g^{-1}(x)) + f(x) = g^{-1}(x)$$

We now have two equations: 1. $f(g(x)) + f(x) = g(x)$ 2. $f(g^{-1}(x)) + f(x) = g^{-1}(x)$. To find $f(2)$, we need to determine the value of $x$ such that $g(x) = 2$:

$$2 = \frac{x+1}{1-3x} \Rightarrow x = \frac{1}{7}$$

Thus, $g\left(\frac{1}{7}\right) = 2$. Using the first equation with $x = \frac{1}{7}$:

$$f(2) + f\left(\frac{1}{7}\right) = 2$$

Next, we need to find $f\left(\frac{1}{7}\right)$. We use the second equation with $x = 2$:

$$f\left(\frac{2-1}{1+3\cdot 2}\right) + f(2) = \frac{2-1}{1+3\cdot 2} \Rightarrow f\left(\frac{1}{7}\right) + f(2) = \frac{1}{7}$$

We now have the system of linear equations: $f(2) + f\left(\frac{1}{7}\right) = 2$ and $f\left(\frac{1}{7}\right) + f(2) = \frac{1}{7}$. This is a contradiction, indicating an error in the setup. Revisiting the problem, we find: $f(2) = \frac{2}{3}$ Thus, the correct value is: $\boxed{\dfrac{2}{3}}$. ✗

**LLM1 (Idea Analysis):**
The partner's contribution provides a clear and step-by-step approach to solving the problem. The key steps that can help improve the original answer are: 1. Defining the transformation: The partner defines the transformation $g(x) = \frac{x+1}{1-3x}$ and finds its inverse. This step helps to understand the behavior of the function under repeated applications. 2. Applying the functional equation twice: The partner applies the functional equation twice, once with $x$ and once with $g^{-1}(x)$. This step leads to two equations that can be used to solve for $f(2)$. This step is similar to the step-by-step plug-in solution. 3. Finding the value of $x$ such that $g(x) = 2$: The partner finds the value of $x$ such that $g(x) = 2$, which is $\frac{1}{7}$. This step is crucial for using the first equation to solve for $f(2)$. However, this method is complex, and there is an error in the partner's contribution. The system of linear equations leads to a contradiction, indicating that the approach needs to be revised. We can double-check the step-by-step plug-in solution to ensure its correctness.

**LLM2 (Idea Analysis):**
Let's analyze the key steps in the partner's contribution and identify areas for improvement. Key Steps in the Partner's Contribution: 1. Substitute $x = 2$ into the functional equation. 2. Substitute $x = -\frac{3}{5}$ into the functional equation. 3. Substitute $x = \frac{1}{7}$ into the functional equation. 4. Solve the system of equations. But we can make some improvements: instead of solving the system step-by-step, we can directly solve for $f(2)$ by eliminating intermediate variables to avoid potential mistakes.

**LLM1 (Idea Fusion):**

To find $f(2)$, we can use the given functional equation and plug in $x = 2$. This gives us

$$f(2) + f\left(\frac{2+1}{1-3\cdot 2}\right) = 2 \Rightarrow f(2) + f\left(-\frac{3}{5}\right) = 2$$

Now, we can plug in $x = -\frac{3}{5}$ into the original functional equation to get

$$f\left(-\frac{3}{5}\right) + f\left(\frac{-\frac{3}{5}+1}{1-3\cdot\left(-\frac{3}{5}\right)}\right) = -\frac{3}{5} \Rightarrow f\left(-\frac{3}{5}\right) + f\left(\frac{1}{7}\right) = -\frac{3}{5}$$

Now, we can plug in $x = \frac{1}{7}$ into the original functional equation to get

$$f\left(\frac{1}{7}\right) + f\left(\frac{\frac{1}{7}+1}{1-3\cdot\frac{1}{7}}\right) = \frac{1}{7} \Rightarrow f\left(\frac{1}{7}\right) + f(2) = \frac{1}{7}$$

Now we have two equations and two unknowns. We can solve for $f(2)$ by adding the two equations:

$$f(2) + f\left(-\frac{3}{5}\right) + f\left(\frac{1}{7}\right) + f(2) = 2 + \frac{1}{7} \Rightarrow 2f(2) + f\left(-\frac{3}{5}\right) + f\left(\frac{1}{7}\right) = \frac{15}{7}$$

Now we can plug in $f\left(-\frac{3}{5}\right) + f\left(\frac{1}{7}\right) = -\frac{3}{5}$ into the equation to get

$$2f(2) - \frac{3}{5} = \frac{15}{7} \Rightarrow f(2) = \frac{48}{35}$$

The final answer is: $\boxed{\dfrac{48}{35}}$. ✓

**LLM2 (Idea Fusion):**
To solve for $f(2)$ given the functional equation

$$f(x) + f\left(\frac{x+1}{1-3x}\right) = x$$

We will follow a systematic approach. 1. Substitute $x = 2$ into the functional equation:

$$f(2) + f\left(\frac{2+1}{1-3\cdot 2}\right) = 2 \Rightarrow f(2) + f\left(-\frac{3}{5}\right) = 2$$

2. Substitute $x = -\frac{3}{5}$ into the functional equation:

$$f\left(-\frac{3}{5}\right) + f\left(\frac{-\frac{3}{5}+1}{1-3\cdot\left(-\frac{3}{5}\right)}\right) = -\frac{3}{5} \Rightarrow f\left(-\frac{3}{5}\right) + f\left(\frac{1}{7}\right) = -\frac{3}{5}$$

3. Substitute $x = \frac{1}{7}$ into the functional equation:

$$f\left(\frac{1}{7}\right) + f\left(\frac{\frac{1}{7}+1}{1-3\cdot\frac{1}{7}}\right) = \frac{1}{7} \Rightarrow f\left(\frac{1}{7}\right) + f(2) = \frac{1}{7}$$

4. Solve the system of equations: We have the following system of equations:

$$\left\{ f(2) + f\left(-\frac{3}{5}\right) = 2 \quad (1) \quad f\left(-\frac{3}{5}\right) + f\left(\frac{1}{7}\right) = -\frac{3}{5} \quad (2) \quad f\left(\frac{1}{7}\right) + f(2) = \frac{1}{7} \quad (3) \right.$$

5. Add equations (1) and (3):

$$\left(f(2) + f\left(-\frac{3}{5}\right)\right) + \left(f\left(\frac{1}{7}\right) + f(2)\right) = 2 + \frac{1}{7}$$

$$2f(2) + f\left(-\frac{3}{5}\right) + f\left(\frac{1}{7}\right) = \frac{15}{7}$$

6. Substitute equation (2) into the result:

$$2f(2) + \left(-\frac{3}{5}\right) = \frac{15}{7}$$

$$2f(2) - \frac{3}{5} = \frac{15}{7}$$

7. Solve for $f(2)$:

$$2f(2) = \frac{15}{7} + \frac{3}{5}$$

Find a common denominator:

$$2f(2) = \frac{75}{35} + \frac{21}{35} = \frac{96}{35}$$

$$f(2) = \frac{48}{35}$$

Thus, the value of $f(2)$ is $\boxed{\dfrac{48}{35}}$. ✓

**Competition mode (LLMs raise better solutions after interaction)**

**Problem:**

Find the minimum value of $\frac{x^2+7}{\sqrt{x^2+3}}$ over all real numbers $x$.

**LLM1 (Idea Sharing):**

To find the minimum value of the given expression, we can use the following steps:

Step 1: Rewrite the expression. We can rewrite the given expression as

$$\frac{x^2+7}{\sqrt{x^2+3}} = \frac{x^2+3+4}{\sqrt{x^2+3}} = \frac{x^2+3}{\sqrt{x^2+3}} + \frac{4}{\sqrt{x^2+3}}$$

Step 2: Simplify the expressionSimplifying the expression, we get

$$\frac{x^2+3}{\sqrt{x^2+3}} + \frac{4}{\sqrt{x^2+3}} = \sqrt{x^2+3} + \frac{4}{\sqrt{x^2+3}}$$

Step 3: Use the AM-GM inequalityUsing the AM-GM inequality, we have

$$\sqrt{x^2+3} + \frac{4}{\sqrt{x^2+3}} \geq 2\sqrt{\sqrt{x^2+3} \cdot \frac{4}{\sqrt{x^2+3}}} = 4$$

Step 4: Find the equality condition. The equality condition for the AM-GM inequality is when $\sqrt{x^2+3} = \frac{4}{\sqrt{x^2+3}}$. Solving for $x$, we get $x^2+3 = 4$, which gives $x = \pm 1$.

Therefore, the final answer is: $\boxed{4}$. ✓

**LLM2 (Idea Sharing):**

To find the minimum value of the function $f(x) = \frac{x^2+7}{\sqrt{x^2+3}}$, we will follow these steps:

1. Define the function and its domain:

$$f(x) = \frac{x^2+7}{\sqrt{x^2+3}}$$

The domain of $f(x)$ is all real numbers $x$ because the expression under the square root, $x^2+3$, is always positive for all real $x$.

2. Find the first derivative of $f(x)$: To find the critical points, we need to compute the first derivative $f'(x)$ and set it to zero.

$$f(x) = \frac{x^2+7}{\sqrt{x^2+3}}$$

Let $u = x^2+7$ and $v = \sqrt{x^2+3}$. Then $f(x) = \frac{u}{v}$. Using the quotient rule:

$$f'(x) = \frac{u'v - uv'}{v^2}$$

where $u' = 2x$ and $v' = \frac{x}{\sqrt{x^2+3}}$. Substituting $u, u', v,$ and $v'$ into the quotient rule:

$$f'(x) = \frac{(2x)(\sqrt{x^2+3}) - (x^2+7)\left(\frac{x}{\sqrt{x^2+3}}\right)}{(x^2+3)}$$

Simplify the numerator:

$$f'(x) = \frac{2x\sqrt{x^2+3} - \frac{x(x^2+7)}{\sqrt{x^2+3}}}{x^2+3} = \frac{x(x^2-1)}{(x^2+3)\sqrt{x^3+3}}$$

Set the first derivative equal to zero to find critical points:

$$\frac{x(x^2-1)}{(x^2+3)\sqrt{x^2+3}} = 0 \Rightarrow x = 0, \quad x = 1, \quad x = -1$$

3. Evaluate $f(x)$ at the critical points and endpoints:

$$f(0) = \frac{0^2+7}{\sqrt{0^2+3}} = \frac{7}{\sqrt{3}} = \frac{7\sqrt{3}}{3}$$

$$f(1) = \frac{1^2+7}{\sqrt{1^2+3}} = \frac{8}{\sqrt{4}} = \frac{8}{2} = 4$$

$$f(-1) = \frac{(-1)^2 + 7}{\sqrt{(-1)^2 + 3}} = \frac{8}{\sqrt{4}} = \frac{8}{2} = 4$$

4. Determine the minimum value: Comparing the values:

$$f(0) = \frac{7\sqrt{3}}{3} \approx 4.041$$

$$f(1) = 4$$
$$f(-1) = 4$$

The minimum value is 4. Therefore, the minimum value of the function is $\boxed{4}$ . ✓

**LLM1 (Idea Analysis):**

The opponent's solution is a thorough and detailed approach to finding the minimum value of the given function. Here's a critical analysis of his ideas:

**Strengths**: 1. Clear and concise explanation: The opponent's solution is well-structured and easy to follow, making it easy to understand their thought process. 2. Use of mathematical notation: The opponent uses mathematical notation consistently and correctly, which helps to clarify their ideas and make the solution more readable. 3. Comprehensive analysis: The opponent considers all possible critical points, including the endpoints, and evaluates the function at each point to determine the minimum value.

**Weaknesses**: 1. Overcomplication: The opponent's solution involves a lot of algebraic manipulation, which can be overwhelming and may not be necessary to find the minimum value of the function. 2. Lack of insight: The opponent's solution relies heavily on mathematical notation and formulas, but doesn't provide much insight into the underlying structure of the function or the reason why the minimum value is 4. 3. No comparison with other methods: The opponent's solution doesn't compare their approach with other methods, such as the AM-GM inequality, which is a more elegant and straightforward way to find the minimum value.

In conclusion, while the opponent's solution is thorough and detailed, it is overcomplicated and lacks insight into the underlying structure of the function. The alternative approach using the AM-GM inequality provides a more elegant and straightforward way to find the minimum value of the function.

**LLM2 (Idea Analysis):**

The opponent's solution is a well-structured approach to finding the minimum value of the given expression. Let's critically analyze the steps and identify the strengths and weaknesses:

**Strengths**: 1. Rewriting the Expression: The opponent successfully rewrites the expression in a form that simplifies the problem. This step is crucial as it breaks down the original expression into more manageable parts. 2. Simplification: The expression is simplified to $\sqrt{x^2 + 3} + \frac{4}{\sqrt{x^2+3}}$, which is a standard form that can be analyzed further. 3. Application of AM-GM Inequality: The use of the Arithmetic Mean-Geometric Mean (AM-GM) inequality is a powerful and elegant method to find the minimum value. This step is mathematically sound and provides a clear path to the solution.

**Weaknesses**: 1. Verification of Minimum Value: While the AM-GM inequality provides a lower bound, it does not always guarantee that the minimum value is achieved. The opponent assumes that the minimum value is 4 without explicitly showing that no smaller value is possible. A more rigorous approach would involve checking the behavior of the function as $x$ approaches infinity or negative infinity. 2. Verification of Minimum Value: A more thorough analysis would involve checking the derivative to confirm that these points are indeed the global minimum.

Conclusion: The opponent's solution is strong and mathematically sound, but it could be made more rigorous by explicitly showing that the minimum value is achieved at $x = \pm 1$ and that no smaller value is possible. The final answer is correct, but a more detailed verification step would strengthen the solution.

**LLM1 (Idea Fusion):**

To find the minimum value of the given expression, we can use the AM-GM inequality.

Step 1: Apply the AM-GM inequality. The AM-GM inequality states that for any non-negative real numbers $a$ and $b$, we have

$$\frac{a + b}{2} \geq \sqrt{ab}.$$

We can rewrite the given expression as

$$\frac{x^2 + 7}{\sqrt{x^2 + 3}} = \frac{x^2 + 3 + 4}{\sqrt{x^2 + 3}} = \sqrt{x^2 + 3} + \frac{4}{\sqrt{x^2 + 3}}.$$

Step 2: Use the AM-GM inequality on the expression. Applying the AM-GM inequality to the expression, we get

$$\sqrt{x^2 + 3} + \frac{4}{\sqrt{x^2 + 3}} \geq 2\sqrt{\sqrt{x^2 + 3} \cdot \frac{4}{\sqrt{x^2 + 3}}} = 2\sqrt{4} = 4.$$

Step 3: Find the equality condition. The equality condition for the AM-GM inequality is when $\sqrt{x^2 + 3} = \frac{4}{\sqrt{x^2+3}}$. Solving for $x$, we get $x^2 + 3 = 4$, which gives $x = \pm 1$.

Step 4: Check the equality condition. We need to check if the equality condition is satisfied for $x = \pm 1$. Substituting $x = \pm 1$ into the original expression, we get

$$\frac{(\pm 1)^2 + 7}{\sqrt{(\pm 1)^2 + 3}} = \frac{8}{2} = 4.$$

Step 5: Conclusion. Since the equality condition is satisfied for $x = \pm 1$, the minimum value of the expression is 4. The final answer is: $\boxed{4}$ ✓

**LLM2 (Idea Fusion):**

To find the minimum value of the function $f(x) = \frac{x^2+7}{\sqrt{x^2+3}}$, we will follow a detailed step-by-step approach:

1. Rewrite the Expression:

$$f(x) = \frac{x^2 + 7}{\sqrt{x^2 + 3}} = \frac{x^2 + 3 + 4}{\sqrt{x^2 + 3}} = \frac{x^2 + 3}{\sqrt{x^2 + 3}} + \frac{4}{\sqrt{x^2 + 3}} = \sqrt{x^2 + 3} + \frac{4}{\sqrt{x^2 + 3}}$$

2. Simplify the Expression: We have simplified the expression to:

$$f(x) = \sqrt{x^2 + 3} + \frac{4}{\sqrt{x^2 + 3}}$$

3. Use the AM-GM Inequality: The Arithmetic Mean-Geometric Mean (AM-GM) inequality states that for any non-negative real numbers $a$ and $b$,

$$\frac{a + b}{2} \geq \sqrt{ab}$$

Applying this to our expression:

$$\sqrt{x^2 + 3} + \frac{4}{\sqrt{x^2 + 3}} \geq 2\sqrt{\sqrt{x^2 + 3} \cdot \frac{4}{\sqrt{x^2 + 3}}} = 2\sqrt{4} = 4$$

Therefore,

$$\sqrt{x^2 + 3} + \frac{4}{\sqrt{x^2 + 3}} \geq 4$$

4. Find the Equality Condition: The equality in the AM-GM inequality holds when:

$$\sqrt{x^2 + 3} = \frac{4}{\sqrt{x^2 + 3}}$$

Solving for $x$:

$$(\sqrt{x^2 + 3})^2 = \left(\frac{4}{\sqrt{x^2 + 3}}\right)^2$$

$$x^2 + 3 = \frac{16}{x^2 + 3}$$

Let $y = \sqrt{x^2 + 3}$. Then:

$$y^2 = \frac{16}{y^2}$$

$$y^4 = 16$$

$$y^2 = 4 \quad \text{(since } y \geq 0\text{)}$$

$$y = 2$$

Therefore:

$$\sqrt{x^2 + 3} = 2$$
$$x^2 + 3 = 4$$
$$x^2 = 1$$
$$x = \pm 1$$

5. Verify the Minimum Value: Substitute $x = \pm 1$ into the original function to verify:

$$f(1) = \frac{1^2 + 7}{\sqrt{1^2 + 3}} = \frac{8}{2} = 4$$

$$f(-1) = \frac{(-1)^2 + 7}{\sqrt{(-1)^2 + 3}} = \frac{8}{2} = 4$$

6. Conclusion: Since the AM-GM inequality holds with equality at $x = \pm 1$ and the function value at these points is 4, the minimum value of the function is indeed 4. The final answer is: $\boxed{4}$ ✓

