# OpenReview forum: "Interactive Learning for LLM Reasoning"
_ICLR.cc/2026/Conference — ICLR 2026 Conference Withdrawn Submission_

### Official Review · Reviewer_JZ8g · 2025-10-20

**Soundness:** 3
**Presentation:** 3
**Contribution:** 3
**Rating:** 4
**Confidence:** 4

**Summary:**

This paper proposes ILR (Interactive Learning for LLM Reasoning), a co-learning framework designed to improve the independent reasoning ability of Large Language Models (LLMs) through training-time multi-agent interaction.
ILR combines Dynamic Interaction, which adaptively switches between cooperative and competitive modes using Item Response Theory (IRT), and Perception Calibration, which integrates peer reward distributions via Group Relative Policy Optimization (GRPO).
Experiments on several reasoning and coding benchmarks show consistent improvements (up to 5%) over single-agent baselines, suggesting that interaction during training can enhance autonomous inference performance.

**Strengths:**

1. The paper explores whether multi-agent interaction can improve the independent reasoning ability of LLMs — a novel and valuable direction beyond traditional system-level optimization.

2. The framework is evaluated across multiple reasoning and coding benchmarks, showing stable and consistent improvements (up to 5%) over strong single-agent baselines.

3. The paper includes insightful ablation and robustness studies, such as comparing cooperative vs. competitive modes and analyzing the effect of dynamic interaction on stronger LLMs.

**Weaknesses:**

1. The estimation of the reasoning ability parameter $\gamma$ is unclear, and the proportion of samples assigned to each strategy (cooperation vs. competition) after IRT-based classification is not reported.
If the estimation of  $\gamma$ is performed only once offline, its reliability may be questionable, as model reasoning ability can evolve throughout training.

2.  While the paper emphasizes adaptive switching between cooperative and competitive strategies, in real human collaboration these two modes often coexist within the same reasoning process. Given that multiple samples are already drawn per question, it would be natural to mix both interaction types within a single batch and compute a unified advantage. Without theoretical or empirical evidence that adaptive switching is superior to parallel coexistence, the contribution of this design choice appears less convincing.

3.  The training procedure involving GRPO and multi-agent interaction is insufficiently detailed. It is unclear whether the interaction phase replaces or complements GRPO rollouts, and how the reward is assigned to individual agents’ actions. If the reward covers the full Idea³ process, it is uncertain how the model achieves autonomous inference afterward. Moreover, the alignment between training and inference prompts is not discussed, raising potential concerns about off-policy distributional shift. Including end-to-end pseudocode for the rollout and GRPO update would significantly improve clarity.

4. Insufficient Motivation and Notation Ambiguity in Equation (5):

   4.1  Should the subscript notation be $l = M  \setminus \{ M_i \}$ ？Is it  $l \in M \setminus \{ M_i \}$ instead?

   4.2 What is the motivation for using the range normalization term $(R_{l,\max} - R_{l,\min})$?

   4.3 The scaling term before the clip function depends on raw differences in magnitude — what proportion of rewards is actually clipped, and could this excessive truncation suppress informative signals?

    4.4 Conceptually, how does incorporating another model’s relative advantage causally enhance the independent reasoning ability of the current agent? The underlying mechanism is unclear.

   4.5 If the formula is adapted from prior work, please provide an explicit citation. Otherwise, conduct an ablation study comparing alternative normalization methods.

5. Reward model implementation not specified.

6. Training cost and cost-effectiveness remain unclear, as the paper lacks quantitative analysis of computational overhead versus performance gains.

7. The use of only two agents provides limited evidence to support the claim of a “multi-agent system.”
In multi-agent interaction settings, both methodological complexity and computational cost increase substantially as the number of agents grows, yet the paper does not evaluate how ILR scales beyond two agents.

8. The paper assumes agents with different backbones, but it is unclear whether the framework would still work with identical models initialized by different prompts or roles.

**Questions:**

See Weakness.

---

> ### Author Response · Authors · 2025-11-24
> **Author Responses (1/3)**
>
> We sincerely appreciate your valuable comment. We are glad that you find our work to be valuable and the ablation studies to be insightful. We address your concerns in detail below:
>
> **Weaknesses:**
>
> ### 1. The estimation of the reasoning ability parameter ...
>
> First, as briefly mentioned in Appendix A.3 of the original version, the initial $\gamma_i$ value for each LLM is measured on a 1000-sample held-out validation set. For instance, the initial $\gamma_i$ values for Llama-3.1-8B-Instruct, Qwen2.5-7B-Instruct, and Qwen2.5-14B-Instruct are 0.59, 0.75, and 0.78, respectively. We also report that the proportion of cooperation/competition across the three groups is 36.30/63.70%, 32.76/67.24%, and 16.87/83.13% for Group 1, Group 2, and Group 3.
>
> Second, we agree that a model's reasoning ability can indeed evolve during training. To investigate this, we conduct an additional experiment to study the impact of dynamically updating $\gamma_i$. Specifically, we re-evaluated $\gamma_i$ periodically every 5 training steps (we have a total of 42 steps). We report the average accuracy on five mathematical evaluation benchmarks and the proportion of cooperation/competition. The results indicate that the performance of periodic updates surpasses that of one-time measurement. As the reasoning capability gap diminishes, the cooperation rate declines, a dynamic that appears more organic, thereby further enhancing performance. We have added this result in **Appendix E.4** of the latest version.
>
> | Model | Avg | periodic-Avg | proportion | periodic-proportion |
> |---|---|---|---|---|
> | Llama-3.1-8B-Instruct | 41.51 | **42.35** ($\uparrow$ 0.84) | 36.30/63.70% | 32.44/67.56% |
> | Qwen2.5-7B-Instruct | 54.44 | **54.82** ($\uparrow$ 0.38) | 36.30/63.70% | 32.44/67.56% |
>
> ### 2. While the paper emphasizes adaptive ...
>
> To directly address your question and provide the empirical evidence you requested, we have conducted a new experiment specifically designed to test this "mixed-types interaction". We modified our sampling procedure as follows: for a given problem, even when our IRT-based mechanism determines to use a cooperation strategy, we enforce a mixture of strategies across the multiple samples. For example, if we generated five distinct rollouts for a single question, we would randomly assign the designated cooperation strategy to 3-5 of these rollouts, while assigning the competition strategy to the remaining rollouts.
>
> As shown in the following table, this shift yielded asymmetric outcomes. Llama-3.1-8B-Instruct achieved a performance gain of 0.40\%, likely because the mixed strategy exposed it to more cooperative reasoning paths on challenging problems. In contrast, Qwen2.5-7B-Instruct experienced a regression of 0.47\%. This decline suggests that the "mixed" samples acted as noise for the stronger model, where forcing it to collaborate (when it should compete) likely constrained its reasoning potential or introduced errors from the weaker partner. We have added the result in **Appendix E.5** of the latest version.
>
> | Model | Avg | mixed-Avg | proportion | mixed-proportion |
> |---|---|---|---|---|
> | Llama-3.1-8B-Instruct | 41.51 | **41.91** ($\uparrow$ 0.40) | 36.30/63.70% | 40.99/59.01% |
> | Qwen2.5-7B-Instruct | **54.44** | 53.97 ($\downarrow$ 0.47) | 36.30/63.70% | 40.99/59.01% |
>
> ### 3. The training procedure ...
>
> The intermediate results within Idea3 are not included in the rollout experience. As mentioned in Section 3.2, Idea3 will generate both an "initial answer" and an "updated answer". Then we will select one of them as the rollout experience based on a predefined rule. Therefore, the rollout only contains a higher-quality step-by-step solution (we can call it "final answer"), and the reward is calculated solely based on this "final answer".
>
> This design choice can ensure:
>
> - **To mitigate the potential prompt shift you mentioned**. Since the rollout only contains higher-quality step-by-step solutions without intermediate results, the LLM is trained on the same type of data that it is expected to produce during inference. Therefore, there is no misalignment between the training and inference prompts.
>
> - **To ensure a fair comparison with single-agent learning baselines**. Including the rich information from the interaction trajectory in the reward signal could introduce a confounding variable, making it difficult to determine whether performance improvements stem from ILR or simply from the additional information in the trajectory.
>
> We have clarified this point in Section 3.2 and added a pseudocode as suggested in **Appendix C** of the latest version.
>
> ### 4.1 Should the subscript ...
>
> Thanks for pointing that out. We have revised this typo in the latest version.

---

> > ### Author Response · Authors · 2025-11-24
> > **Author Responses (2/3)**
> >
> > ### 4.2 What is the motivation ...
> >
> > This normalization makes the resulting reward shaping signal invariant to the absolute scale of the reward model. For example, consider a raw reward difference of 0.5 between two LLMs’ outputs. If the reward model operates on a scale of 1, a difference of 0.5 is extremely significant, representing 50% of the entire possible reward range. Conversely, if the reward model operates on a scale of 5, the same absolute difference of 0.5 is far less significant, representing only 10% of the total range. Without normalization, the reward shaping term would be highly sensitive to the specific output range of the reward model, and the RL training could be unstable.
> >
> >
> > ### 4.3 The scaling term ...
> >
> > The clipping range is [-1, 1] in our two-agent setting, which can filter out extreme noise that could otherwise destabilize the training process. To directly address your question about the practical effect of this mechanism, we report the proportion of rewards that are actually clipped based on Group1. It shows that the percentage of instances where the normalized reward difference exceeded the boundary is quite small. Therefore, we believe that this clip will not overly suppress the informative signals. We have added the proportion in **Appendix A.4** of the latest version.
> >
> > |  | Group1 | Group2 | Group3 |
> > |---|---|---|---|
> > | Llama-3.1-8B-Instruct | 5.04% | 7.15% | - |
> > | Qwen2.5-7B-Instruct | 2.19% | - | 1.64% |
> > | Qwen2.5-14B-Instruct | - | 3.59% | 2.34% |
> >
> >
> > ### 4.4 Conceptually, how does incorporating ...
> >
> > MAPoRL [1] has provided a game-theoretic perspective on how one agent's answer influences another's. They have also empirically demonstrated that reward shaping, i.e., modifying an agent's reward based on the outcomes of others, is an effective technique for promoting beneficial multi-agent learning.
> >
> > However, current reward shaping relies on the manual addition of predefined hyperparameter signals to the rewards. This approach can result in discrete and coarse-grained reward signals, which in turn limit the scalability of reward shaping. We have advanced this method to obviate the need for manual design and yield a more nuanced and fine-grained incentive structure.
> >
> > [1] Park C, Han S, Guo X, et al. Maporl: Multi-agent post-co-training for collaborative large language models with reinforcement learning[C]//Proceedings of the 63rd Annual Meeting of the Association for Computational Linguistics (Volume 1: Long Papers). 2025: 30215-30248.
> >
> > ### 4.5 If the formula is adapted ...
> >
> > The formula 5 is an innovative way we proposed to produce an automatic reward incentive. As you suggested, we have run z-score normalization (mark as z-), and we report the average accuracy on five mathematical evaluation benchmarks and the clip ratio. The results indicated that implementing z-score normalization within the ILR framework yields inferior performance compared to our method. We attribute this degradation to the aggressive clipping associated with z-score normalization, which tends to suppress the magnitude of incentive signals. This observation further corroborates the superiority of our adopted min-max normalization strategy. We have added the results in **Appendix E.1** of the latest version.
> >
> > | Model | Avg | z-Avg | ratio | z-ratio |
> > |---|---|---|---|---|
> > | Llama-3.1-8B-Instruct | **41.51** | 40.74 ($\downarrow$ 0.77) | 5.04\% | 20.45\% |
> > | Qwen2.5-7B-Instruct | **54.44** | 53.94 ($\downarrow$ 0.50) | 2.19\% | 9.10\% |
> >
> > ### 5. Reward model implementation not specified.
> >
> > We have mentioned the reward model we used in Appendix A.2 of the original version, i.e., we utilize Llama-3-8b-rm-mixture as the reward model to rate sampled answers. For clarity, we have further mentioned this point in **Section 4.1** of the latest version.

---

> > > ### Author Response · Authors · 2025-11-24
> > > **Author Responses (3/3)**
> > >
> > > ### 6. Training cost and cost-effectiveness ...
> > >
> > > Thanks for the suggestion. We compare the training time and performance of GRPO and ILR here and report the average accuracy on five mathematical evaluation benchmarks (\%, without []) and the training time (hours, within []).
> > >
> > > Note that the training durations reported for ILR-G$i$ represent the aggregated time required for both LLMs within the group. For instance, in the ILR-G1 setting, the combined training time for Llama-3.1-8B-Instruct and Qwen2.5-7B-Instruct amounts to 8 hours. In comparison, training these models individually using standard GRPO requires a total of 7 hours (3.75 hours for Llama-3.1-8B-Instruct and 3.25 hours for Qwen2.5-7B-Instruct). Consequently, incorporating ILR introduces only a marginal computational overhead of one hour. This slight increase in cost is justified by the performance gains, yielding improvements of approximately 3% for Llama-3.1-8B-Instruct and 1% for Qwen2.5-7B-Instruct. We believe this cost increase is acceptable and have added the result in **Appendix E.6** of the latest version.
> > >
> > > | Model | GRPO | ILR-G1 | ILR-G2 | ILR-G3 |
> > > |---|---|---|---|---|
> > > | Llama-3.1-8B-Instruct | 38.39[3.75] | 41.51[8.00] | 41.10[12.00] | - |
> > > | Qwen2.5-7B-Instruct | 53.58[3.25] | 54.44[8.00] | - | 54.59[10.00] |
> > > | Qwen2.5-14B-Instruct | 57.55[5.50] | - | 58.95[12.00] | 59.30[10.00] |
> > >
> > >
> > > ### 7. The use of only two agents ...
> > >
> > > We agree completely that testing ILR with more agents is a meaningful direction to analyze its scaling trends. So we have conducted a new experiment involving three agents (ILR-G4): Llama-3.1-8B-Instruct, Qwen2.5-7B-Instruct, and Qwen2.5-14B-Instruct. We report the average accuracy on five mathematical evaluation benchmarks, and the result indicates that ILR continues to be effective in this expanded setting, demonstrating a positive scaling trend. We have included the result in **Section 4.2.4** of the latest version.
> > >
> > > | Model | Base | ILR-G1 | ILR-G2 | ILR-G3 | ILR-G4 |
> > > |---|---|---|---|---|---|
> > > | Llama-3.1-8B-Instruct | 34.15 | 41.51 | 41.10 | - | **42.48** |
> > > | Qwen2.5-7B-Instruct | 51.76 | 54.44 | - | 54.59 | **55.12** |
> > > | Qwen2.5-14B-Instruct | 55.57 | - | 58.95 | 59.30 | **59.95** |
> > >
> > > ### 8. The paper assumes agents ...
> > >
> > > To directly investigate the scenario you proposed, we conduct a new experiment, where we apply ILR to the identical Llama-3.1-8B-Instruct with two different input prompts.
> > >
> > >
> > > **Prompt1**: ${Question.}$ Please reason step by step.
> > >
> > > **Prompt2**: ${Question}.$ Please double-check the question, first extract the keywords and think about the knowledge about these keywords, then solve the question.
> > >
> > >
> > > Two notable findings emerge from this analysis. **First**, while ILR with an identical LLM still yields performance improvements over single-agent baselines, it does not surpass the performance of ILR with different LLMs. This suggests that while diversity can be simulated through varied prompting, distinct models provide a richer and more effective source of alternative perspectives. The interaction between truly heterogeneous agents appears to generate a more powerful learning signal. **Second**, the post-training performance of the Llama model utilizing Prompt 2 was superior to its counterpart. This observation underscores that the granularity of instruction prompts plays a significant role in influencing the outcomes of interactive learning. We have added the result in **Appendix E.7** of the latest version.
> > >
> > > |  | GSM8K | MATH-500 | Minerva Math | Olympiad Bench | AIME24&25 | Avg |
> > > |---|---|---|---|---|---|---|
> > > | Llama-3.1-8B-Instruct | 82.87 | 49.80 | 22.79 | 13.63 | 1.67 | 34.15 |
> > > | Prompt1 | 87.79 | 54.20 | 29.41 | 20.30 | 6.67 | 39.67 |
> > > | Prompt2 | 87.72 | 55.60 | 30.51 | 20.15 | 6.67 | 40.13 |
> > > | ILR-Group1 | **89.39** | **55.80** | 30.15 | 22.22 | **10.00** | **41.51** |
> > > | ILR-Group2 | 87.26 | 55.20 | **33.82** | **22.52** | 6.67 | 41.10 |

---

> ### Author Response · Authors · 2025-11-27
> **Looking forward to your feedback**
>
> Dear Reviewer JZ8g,
>
> We are eager to ensure that we have adequately addressed your concerns and are prepared to offer further clarifications or address any additional questions you may have.
>
> We would like to express our heartfelt gratitude for the time and effort you have dedicated to reviewing our work. It has been a pleasure to engage with you throughout this process.
>
> Best regards,
>
> The authors

---

> ### Author Response · Authors · 2025-11-28
> **A summary of our revision**
>
> Dear Reviewer JZ8g,
>
> Thanks for your valuable comments, and we would be grateful if you could allocate some time to review our revision.
>
> We understand that you have a multitude of responsibilities. To facilitate a swift evaluation of our revisions, we have summarized the corresponding changes as follows:
>
> **Periodic update of model reasoning ability**: We have added the discussion of periodic update of reasoning ability $\gamma_i$ and results in **Appendix E.4**.
>
> **Mixed-type interaction**: We have added the discussion of mixed-type interaction within the rollout group and results in **Appendix E.5**.
>
> **Training procedure**: We have clarified the trajectory collection in **Section 3.2** and added a pseudocode as suggested in **Appendix C**.
>
> **About formula 5**:  We have clarified the motivation of min-max normalization, reported the clip ratio, elaborated on the reward shaping, and the comparison of z-score normalization in **Appendix A.4** and **Appendix E.1**.
>
> **Cost Table**: We have added a cost table and discussed the cost-performance tradeoff in **Appendix E.6**.
>
> **ILR with three LLMs**: We have included the result of ILR with three LLMs in **Section 4.2.4**, which demonstrates a positive scaling trend.
>
> **Same LLM with different input prompts**: We have applied ILR to the identical Llama-3.1-8B-Instruct with two different input prompts and added the result in **Appendix E.7**.
>
> Please let us know if you have any additional concerns or questions. We kindly request your feedback on whether our responses have satisfactorily addressed your concerns. Thank you for your time and consideration!
>
> Best regards,
>
> The authors

---

### Official Review · Reviewer_opn6 · 2025-10-31

**Soundness:** 2
**Presentation:** 3
**Contribution:** 3
**Rating:** 4
**Confidence:** 3

**Summary:**

The paper investigates whether multi-agent interaction can enhance a single LLM’s independent problem-solving ability. The proposed ILR framework comprises two components: Dynamic Interaction and Perception Calibration. In Dynamic Interaction, the strategy (cooperation vs. competition) is adaptively selected based on problem difficulty and model capability. The LLMs then communicate via Idea3 (idea sharing, idea analysis, and idea fusion). In Perception Calibration, ILR adjusts rewards using the collaborator LLMs’ reward distribution features and trains the LLMs with GRPO. Evaluations on five benchmarks, compared against single-agent and multi-agent fine-tuning methods, demonstrate the effectiveness of ILR.

**Strengths:**

1. This work proposes the problem of enhancing a single agent’s capability via multi-agent collaboration and designs a novel learning framework named ILR.
2. The Dynamic Interaction module automatically chooses cooperation or competition by estimating the problem difficulty and model capability, avoiding the cost of manually setting a cooperation ratio. Ablations show that pure cooperation or pure competition is suboptimal, while dynamic selection is more robust.
3. By using collaborators’ reward distribution features (max/min/avg) to continuously shape the agent’s own reward and optimizing the single agent with GRPO, the method achieves strong experimental performance.

**Weaknesses:**

1. The motivation for Dynamic Interaction—“hard problems benefit from cooperation, easy problems from competition,” inspired by classroom observations—may not hold across all task settings. More conditioning variables may need to be incorporated to improve robustness.

2. The paper builds on GRPO with multi-agent discussion to optimize training but does not compare against single-agent GRPO variants such as [1] and [2]. What is the essential advantage of introducing multi-agent enhancement over improved single-agent GRPO baselines?

   [1] CPPO: Accelerating the training of group relative policy optimization-based reasoning models.

   [2] COG-RETHINKER: Hierarchical Metacognitive Reinforcement Learning for LLM Reasoning.

3. Using MATH-500 as an example, untrained collaborative inference already boosts Llama-3.1-8B from 49.80 to 64.00/66.20 (Table 4 Debate), whereas after ILR training the single-agent score on this benchmark rises only to 55.80/55.20 (Table 1). This “multi-agent at training, single agent at inference” setup may underutilize the potential advantages and ceiling of multi-agent systems at deployment.

**Questions:**

1. Training cost vs. single-agent learning methods: Is the performance gain partly due to more sampling? In ILR, each question involves multiple agents generating sampled answers, which could increase trajectories and reward evaluations relative to single-agent training.
2. Experiments are limited to two agents. Does increasing the number of agents in the discussion further improve training effectiveness?

**Details Of Ethics Concerns:**

The paper raises no ethical concerns.

---

> ### Author Response · Authors · 2025-11-24
> **Author Responses**
>
> We sincerely appreciate your valuable comment. We are glad that you find our work to be novel. We address your concerns in detail below:
>
> **Weaknesses:**
>
> ### 1. The motivation for Dynamic Interaction ...
>
> This is an insightful suggestion. For many common reasoning tasks, such as mathematics and standard question-answering, the intrinsic difficulty of a problem is one of the most direct and dominant factors influencing whether an individual chooses to persevere independently or seek collaborative assistance. However, in scenarios like creative writing, the "problem difficulty" does not necessarily correlate directly with the quality of the final output. Even a "simple problem" might benefit more from a cooperative strategy, as this could foster a richer blend of stylistic diversity by combining the unique perspectives of different LLMs. In such cases, the goal is not just to find a correct solution but to generate a high-quality, creative artifact.
>
> Therefore, for such specific task settings, we need to incorporate additional conditioning variables to improve the robustness of Dynamic Interaction. We have discussed this point in **Appendix D** of the latest version.
>
> ### 2. The paper builds on GRPO ...
>
> We agree completely that comparing ILR against advanced single-agent GRPO variants is crucial to isolate and demonstrate the specific benefits of interactive learning. We have run CPPO on Llama-3.1-8B-Instruct, and the result clearly indicates that our ILR framework achieves superior performance compared to such an advanced GRPO variant. We have integrated the results into **Appendix E.3** of the latest version.
>
> |  | GSM8K | MATH-500 | Minerva Math | Olympiad Bench | AIME24&25 | Avg |
> |---|---|---|---|---|---|---|
> |Llama-3.1-8B-Instruct|82.87|49.80|22.79|13.63|1.67|34.15|
> |CPPO|84.84|54.60|27.57|18.85|6.67|38.51|
> |ILR-Group1|**89.39**|**55.80**|30.15|22.22|**10.00** | **41.51** |
> |ILR-Group2|87.26|55.20|**33.82**|**22.52**|6.67|41.10|
>
> ### 3. Using MATH-500 as an example ...
>
> The "multi-agent at training, single agent at inference" setup isolates the effect of other LLMs at the inference stage, which allows us to explore whether an LLM can become a better independent reasoner after ILR training.
>
> Meanwhile, we want to emphasize that our ILR is not in conflict with multi-agent inference. We conduct a new experiment, where we deploy ILR-trained LLMs in a multi-agent inference setting (mark as ILR-), and the results show that the ILR-trained LLMs achieve better performance than the untrained LLMs in the same multi-agent inference setup. It indicates that ILR training does not diminish an agent's ability to collaborate effectively at deployment. We have added these results to **Table 4** in the latest version.
>
> |  | Llama-8B |  | Qwen-7B |  | Qwen-14B |  |
> |---|---|---|---|---|---|---|
> | Inference Paradigm | Group1 | Group2 | Group1 | Group3 | Group2 | Group3 |
> | Debate | 64.00 | 66.20 | 74.60 | 80.00 | 79.20 | 81.00 |
> | ILR-Debate | 65.60 $\uparrow$ | 71.20 $\uparrow$ | 81.20 $\uparrow$| 81.00 $\uparrow$ | 79.40 $\uparrow$ | 81.20 $\uparrow$ |
> | Idea3 | 63.40 | 62.00 | 75.60 | 77.80 | 79.80 | 82.00 |
> | ILR-Idea3 | 65.00 $\uparrow$ | 67.80 $\uparrow$ | 76.00 $\uparrow$ | 78.80 $\uparrow$ | 80.40 $\uparrow$ | 82.40 $\uparrow$ |
>
> **Questions:**
>
> ### 1. Training cost ...
>
> To address this concern directly and empirically, we conduct a new experiment on Llama-3.1-8B-Instruct, where we double the rollout number of the single-agent GRPO baseline to isolate the effect of more sampling (2-GRPO). The results show that ILR still achieves superior performance, indicating that the performance gain is not coming from more sampling. We have added the results in **Appendix E.3** of the latest version.
>
> |  | GSM8K | MATH-500 | Minerva Math | Olympiad Bench | AIME24&25 | Avg |
> |---|---|---|---|---|---|---|
> | Llama-3.1-8B-Instruct | 82.87 | 49.80 | 22.79 | 13.63 | 1.67 | 34.15 |
> | 2-GRPO | 86.43 | 53.20 | 28.31 | 18.22 | 6.67 | 38.57 |
> | ILR-Group1 | **89.39** | **55.80** | 30.15 | 22.22 | **10.00** | **41.51** |
> | ILR-Group2 | 87.26 | 55.20 | **33.82** | **22.52** | 6.67 | 41.10 |
>
> ### 2. Experiments are limited to two agents ...
>
> We agree completely that testing ILR with more agents is a meaningful direction to analyze its scaling trends. So we have conducted a new experiment involving three agents (ILR-G4): Llama-3.1-8B-Instruct, Qwen2.5-7B-Instruct, and Qwen2.5-14B-Instruct. We report the average accuracy on five mathematical evaluation benchmarks, and the result indicates that ILR continues to be effective in this expanded setting, demonstrating a positive scaling trend. We have included the result in **Section 4.2.4** of the latest version.
>
> | Model | Base | ILR-G1 | ILR-G2 | ILR-G3 | ILR-G4 |
> |---|---|---|---|---|---|
> | Llama-3.1-8B-Instruct | 34.15 | 41.51 | 41.10 | - | **42.48** |
> | Qwen2.5-7B-Instruct | 51.76 | 54.44 | - | 54.59 | **55.12** |
> | Qwen2.5-14B-Instruct | 55.57 | - | 58.95 | 59.30 | **59.95** |

---

> ### Author Response · Authors · 2025-11-27
> **Looking forward to your feedback**
>
> Dear Reviewer opn6,
>
> We are eager to ensure that we have adequately addressed your concerns and are prepared to offer further clarifications or address any additional questions you may have.
>
> We would like to express our heartfelt gratitude for the time and effort you have dedicated to reviewing our work. It has been a pleasure to engage with you throughout this process.
>
> Best regards,
>
> The authors

---

> ### Author Response · Authors · 2025-11-28
> **A summary of our revision**
>
> Dear Reviewer opn6,
>
> Thanks for your valuable comments, and we would be grateful if you could allocate some time to review our revision.
>
> We understand that you have a multitude of responsibilities. To facilitate a swift evaluation of our revisions, we have summarized the corresponding changes as follows:
>
> **Dynamic Interaction**: We have discussed the robustness of Dynamic Interactions in **Appendix D**.
>
> **Baseline**: We have discussed advanced GRPO variants, and added the results of CPPO in **Appendix E.3**.
>
> **Multi-agent Inference**: We show that ILR training does not diminish an agent's ability to collaborate effectively at deployment. We have added these results to **Table 4**.
>
> **Problem of more sampling**:  We double the rollout number of the single-agent GRPO baseline to isolate the effect of more sampling (2-GRPO). We have added the results in **Appendix E.3**.
>
> **ILR with three LLMs**: We have included the result of ILR with three LLMs in **Section 4.2.4**, which demonstrates a positive scaling trend.
>
> Please let us know if you have any additional concerns or questions. We kindly request your feedback on whether our responses have satisfactorily addressed your concerns. Thank you for your time and consideration!
>
> Best regards,
>
> The authors

---

### Official Review · Reviewer_bbVL · 2025-11-02

**Soundness:** 3
**Presentation:** 3
**Contribution:** 2
**Rating:** 4
**Confidence:** 4

**Summary:**

This paper proposes a multi-agent learning framework that aims to enhance individual LLMs' reasoning capabilities through interaction, rather than building collaborative multi-agent systems for deployment. The key innovation lies in two components: (1) Dynamic Interaction that adaptively chooses between cooperation and competition based on question difficulty (estimated via self-ranking and Item Response Theory), followed by a communication paradigm (Idea Sharing, Analysis, Fusion), and (2) Perception Calibration that automatically incorporates peer reward distributions into each agent's reward function using GRPO. Experiments on Llama-3.1-8B and Qwen2.5-7B/14B across math and coding benchmarks show consistent improvements of up to 5% over single-agent baselines.

**Strengths:**

- The distinction between training for system-level vs. individual-level improvement is thoughtful and practically relevant. The paper correctly identifies that existing multi-agent work (MALT, ReMA, MAPoRL) focuses on building collaborative systems that require multiple agents at inference, which differs from human learning, where individuals improve through interaction but solve problems independently afterward.


- Using self-ranking combined with Item Response Theory to dynamically determine cooperation vs. competition is theoretically grounded. The IRT formulation with the 1.7 coefficient is well-established in educational assessment. The ablation study in Figure 3 effectively demonstrates that dynamic interaction outperforms pure cooperation (p=1.0) or pure competition (p=0.0), validating this design choice.


- The evaluation across multiple model families (Llama and Qwen), scales (7B-14B), and benchmarks (5 math + 1 code) is thorough

**Weaknesses:**

- While the combination is novel, the individual pieces are incremental. The Idea³ framework is essentially a structured variant of existing debate/reflection paradigms. Self-ranking for difficulty estimation and Reward shaping in multi-agent RL have been explored.


- The "up to 5%" improvement claim appears only for Llama-3.1-8B on one metric (average across benchmarks). Most models show 1-2% gains, and on individual benchmarks, improvements are often within noise. I feel there is some overclaim about results. The AIME results look impressive (e.g., 1.67→10.00 for Llama) but these are on tiny test sets with high variance. The paper doesn't report confidence intervals or significance tests


- The paper claims to be "the first to investigate whether multi-agent learning can more effectively enhance an LLM's independent reasoning capability" but doesn't compare against the extensive literature on self-reflection and self-improvement[1,2, 3, 4]


[1] Renze, M., & Guven, E. (2024). Self-Reflection in LLM Agents: Effects on Problem-Solving Performance. arXiv:2405.06682

[2] Shinn, N., Cassano, F., Labash, B., Gopinath, A., Narasimhan, K., & Yao, S. (2023). Reflexion: Language Agents with Verbal Reinforcement Learning. arXiv:2303.11366

[3] Dou, Z., Yang, C., Wu, X., Chang, K., & Peng, N. (2024). Re-ReST: Reflection-Reinforced Self-Training for Language Agents. In Proceedings of EMNLP 2024.

[4] Zhang, W., Shen, Y., Wu, L., Peng, Q., Wang, J., Zhuang, Y., & Lu, W. (2024). Self-Contrast: Better Reflection Through Inconsistent Solving Perspectives. In Proceedings of ACL 2024, pp. 3602-3622.

**Questions:**

Why is IRT the right framework here? The assumption that P(solve) follows a logistic function seems arbitrary for LLMs. Did you try simpler heuristics like directly using self-ranking scores?


Did you try ILR with more than 2 agents? With heterogeneous role assignments (like MALT's generator/verifier/refiner)?


Can you provide confidence intervals, especially for low-data benchmarks like AIME?

---

> ### Author Response · Authors · 2025-11-24
> **Author Responses (1/2)**
>
> We sincerely appreciate your valuable comment. We are glad that you find our work to be novel and the experiment to be thorough. We address your concerns in detail below:
>
> **Weaknesses:**
>
> ### 1. While the combination is novel ...
>
> We agree that the individual elements have been explored in previous research. However, we believe our contribution lies not only in the novel integration of these elements but also in the advanced refinements we have made to the individual pieces.
>
> **First**, the Idea3 framework is more than a structured variant of existing debate or reflection paradigms. Many contemporary debate frameworks for LLMs focus on generating a better answer by having one LLM take another's output as advice. Such direct debate can inadvertently encourage a weaker LLM to simply adopt the answer of a stronger one, bypassing the critical thinking process. However, our Idea3 is designed to foster critical thought within the LLM communication, which is a novel aspect not present in the current LLM debate/reflection.
>
> **Second**, current reward shaping relies on the manual addition of predefined hyperparameter signals to the rewards. This approach can result in discrete and coarse-grained reward signals, which in turn limit the scalability of reward shaping. We have advanced this method to obviate the need for manual design and yield a more nuanced and fine-grained incentive structure.
>
> **Finally**, we would like to re-emphasize the core contribution of our work, i.e., the deep coupling of these refined components. By doing so, we aim to more accurately simulate and study the human-like high-level behaviors in existing LLMs within a context that mirrors real-world human learning scenarios.
>
> We have clarified these points in **Section 3** of the latest version.
>
>
> ### 2. The "up to 5%" improvement ...
>
> Thanks for the suggestion. We have run the evaluation with four other random seeds, and updated the mean value (**Table 1**) and 95\% confidence intervals (**Table 8**) in the latest version. Here we present the AIME24&25 results for reference, and the results are aligned with our findings.
>
> | Llama-3.1-8B-Instruct | 2.33 $\pm$ 1.13 | Qwen2.5-7B-Instruct | 9.33 $\pm$ 2.78  | Qwen2.5-14B-Instruct | 12.33 $\pm$ 3.76 |
> | --- | --- | --- | --- | --- | --- |
> | SFT                   | 3.00 $\pm$ 0.92 | SFT                 | 11.33 $\pm$ 3.70 | SFT                  | 13.00 $\pm$ 3.07 |
> | DPO                   | 2.67 $\pm$ 1.13 | DPO                 | 10.33 $\pm$ 3.40 | DPO                  | 14.33 $\pm$ 1.14 |
> | PPO                   | 4.00 $\pm$ 2.36 | PPO                 | 11.67 $\pm$ 2.07 | PPO                  | 16.67 $\pm$ 2.53 |
> | GRPO                  | 4.00 $\pm$ 2.36 | GRPO                | 13.00 $\pm$ 2.70 | GRPO                 | 16.00 $\pm$ 3.14 |
> | Reinforce++           | 3.33 $\pm$ 2.27 | Reinforce++         | 12.67 $\pm$ 1.85 | Reinforce++          | 16.67 $\pm$ 4.86 |
> | DebateFT-Group1       | 2.67 $\pm$ 1.85 | DebateFT-Group1     | 10.33 $\pm$ 0.93 | DebateFT-Group2      | 13.67 $\pm$ 1.73 |
> | DebateFT-Group2       | 2.67 $\pm$ 1.85 | DebateFT-Group3     | 12.33 $\pm$ 1.85 | DebateFT-Group3      | 12.00 $\pm$ 3.70 |
> | ILR-Group1            | **9.00** $\pm$ 1.14  | ILR-Group1          | 16.33 $\pm$ 2.26 | ILR-Group2           | **20.00** $\pm$ 2.53 |
> | ILR-Group2            | 6.67 $\pm$ 1.46  | ILR-Group3          | **17.00** $\pm$ 2.26 | ILR-Group3           | 19.66 $\pm$ 2.70 |
>
> ### 3. The paper claims to be ...
>
> Thank you for bringing to our attention the extensive and important literature on self-reflection and self-improvement. We agree that a detailed discussion of this related work is crucial for positioning our contribution accurately.
>
> We would like to clarify the core distinction between our ILR and the self-reflection/self-improvement paradigm. Self-reflection and self-improvement represent a significant direction in enhancing agent capabilities through an intra-agent feedback loop. The central idea is that an agent improves by analyzing its own generated trajectories, answers, or reasoning processes. In contrast, the core idea of our ILR is to foster critical thinking and improvement through structured interaction with an external peer. We have run Re-ReST on Llama-3.1-8B-Instruct, and our results indicate that the ILR framework demonstrates superior performance.
>
> We have discussed self-reflection single-agent learning, and added the result in **Appendix E.3** of the latest version.
>
> |  | GSM8K | MATH-500 | Minerva Math | Olympiad Bench | AIME24&25 | Avg |
> |---|---|---|---|---|---|---|
> | Llama-3.1-8B-Instruct | 82.87 | 49.80 | 22.79 | 13.63 | 1.67 | 34.15 |
> | Re-ReST | 84.23 | 49.80 | 29.04 | 15.56 | 5.00 | 36.73 |
> | ILR-Group1 | **89.39** | **55.80** | 30.15 | 22.22 | **10.00** | **41.51** |
> | ILR-Group2 | 87.26 | 55.20 | **33.82** | **22.52** | 6.67 | 41.10 |

---

> > ### Author Response · Authors · 2025-11-24
> > **Author Responses (2/2)**
> >
> > **Questions:**
> >
> > ### 1. Why is IRT ...
> >
> > - Theoretical Appropriateness of IRT.
> >
> > We chose IRT because it offers a principled and theoretical foundation for our scenario. IRT is a mature framework from educational measurement designed to simultaneously model the relationship between three key variables: the ability of an individual, the difficulty of a question, and the probability of the individual correctly solving the question. This is highly analogous to our scenario: we aim to dynamically select an interaction strategy based on the LLM's ability and the problem's difficulty. Furthermore, previous works have investigated the intersection of IRT and AI/LLM [1, 2, 3], demonstrating its relevance and utility in our domain. We have discussed this point in **Section 3.2** of the latest version.
> >
> > [1] Wang X, Jiang L, Hernandez-Orallo J, et al. Evaluating general-purpose AI with psychometrics[J]. arXiv preprint arXiv:2310.16379, 2023.
> >
> > [2] Lalor J P, Rodriguez P, Sedoc J, et al. Item response theory for natural language processing[C]//Proceedings of the 18th Conference of the European Chapter of the Association for Computational Linguistics: Tutorial Abstracts. 2024: 9-13.
> >
> > [3] Zhuang Y, Liu Q, Ning Y, et al. Efficiently measuring the cognitive ability of llms: An adaptive testing perspective[J]. 2023.
> >
> > - Comparison with Simpler Heuristics.
> >
> > As illustrated in **Section 4.2.3**, we directly use the self-ranked difficulty scores and manually defined threshold to determine the proportion of competitive versus cooperative interactions.
> >
> > Our findings, presented in **Figure 3**, show that although IRT is not always optimal, it achieves competitive results while eliminating manual intervention. This demonstrates the feasibility of IRT as a principled mechanism for integrating problem difficulty with LLM reasoning abilities.
> >
> >
> > ### 2. Did you try ILR with more than...
> >
> > - More than 2 agents.
> >
> > We agree completely that testing ILR with more agents is a meaningful direction to analyze its scaling trends. So we have conducted a new experiment involving three agents (ILR-G4): Llama-3.1-8B-Instruct, Qwen2.5-7B-Instruct, and Qwen2.5-14B-Instruct. We report the average accuracy on five mathematical evaluation benchmarks, and the result indicates that ILR continues to be effective in this expanded setting, demonstrating a positive scaling trend. We have included the result in **Section 4.2.4** of the latest version.
> >
> > | Model | Base | ILR-G1 | ILR-G2 | ILR-G3 | ILR-G4 |
> > |---|---|---|---|---|---|
> > | Llama-3.1-8B-Instruct | 34.15 | 41.51 | 41.10 | - | **42.48** |
> > | Qwen2.5-7B-Instruct | 51.76 | 54.44 | - | 54.59 | **55.12** |
> > | Qwen2.5-14B-Instruct | 55.57 | - | 58.95 | 59.30 | **59.95** |
> >
> >
> > - On Heterogeneous Role Assignments.
> >
> > Our ILR intentionally employs a homogeneous design where all agents are problem solvers. This choice is directly tied to our core research question: to investigate whether and how interaction within a multi-agent environment can enhance the independent problem-solving capabilities of each individual LLM. In this context, all agents share the same role and learn from each other through interaction to achieve mutual improvement.
> >
> > However, we believe the idea of incorporating heterogeneous roles is highly inspiring. An exciting future direction would be to design a hybrid system. For example, one could have multiple "generator" agents that are trained internally via ILR to enhance their respective independent generation capabilities. Concurrently, the system could include one or more "verifier" or "refiner" agents responsible for evaluating and integrating the outputs from these ILR-strengthened generators. Such a hybrid model could potentially lead to an even more complex, robust, and powerful multi-agent system. We find this to be a very interesting point for future research and have incorporated this discussion into **Appendix D** of the latest version.
> >
> > ### 3. Can you provide confidence intervals ...
> >
> > Thanks for the suggestion. We have run the evaluation with four other random seeds, and updated the mean value (**Table 1**) and 95\% confidence intervals (**Table 8**) in the latest version. We also present the AIME24&25 results for reference in the response for Weakness 2.

---

> ### Author Response · Authors · 2025-11-27
> **Looking forward to your feedback**
>
> Dear Reviewer bbVL,
>
> We are eager to ensure that we have adequately addressed your concerns and are prepared to offer further clarifications or address any additional questions you may have.
>
> We would like to express our heartfelt gratitude for the time and effort you have dedicated to reviewing our work. It has been a pleasure to engage with you throughout this process.
>
> Best regards,
>
> The authors

---

> ### Author Response · Authors · 2025-11-28
> **A summary of our revision**
>
> Dear Reviewer bbVL,
>
> Thanks for your valuable comments, and we would be grateful if you could allocate some time to review our revision.
>
> We understand that you have a multitude of responsibilities. To facilitate a swift evaluation of our revisions, we have summarized the corresponding changes as follows:
>
> **Clarification about individual component**: We elaborate on the Appropriateness of IRT and the advanced refinements we have made to the individual pieces in **Section 3**.
>
> **Confidence intervals**: We have run the evaluation with four other random seeds, and updated the mean value (**Table 1**) and 95% confidence intervals (**Table 8**).
>
> **Baseline**: We have discussed self-reflection single-agent learning, and added the results of Re-ReST in **Appendix E.3**.
>
> **ILR with three LLMs**: We have included the result of ILR with three LLMs in **Section 4.2.4**, which demonstrates a positive scaling trend.
>
> Please let us know if you have any additional concerns or questions. We kindly request your feedback on whether our responses have satisfactorily addressed your concerns. Thank you for your time and consideration!
>
> Best regards,
>
> The authors

---

### Author Response · Authors · 2025-11-24
**General Response to All Reviewers**

Dear Reviewers,

We sincerely appreciate the reviewers' time and effort in evaluating our paper and their thoughtful feedback. We are pleased that the reviewers recognize the significance of our work on Multi-agent system Reinforcement Learning (MARL) [```bbVL```, ```opn6```, ```JZ8g```], the effectiveness of our ILR [```bbVL```, ```opn6```, ```JZ8g```], and the well-designed ablation studies [```JZ8g```]. We have made every effort to faithfully address your comments in the responses and have uploaded the updated manuscript. As suggested by the reviewers, we summarize the key updates and new experiments included in this revision:

**1. Clarified Individual Components within ILR:** We distinguish Idea3 and Reward shaping with existing techniques, elaborate the theoretical appropriateness of IRT, clarify the rollout experience collection process, provide the clip ratio of formula 5, and add the pseudocode of ILR in **Section 3**, **Appendix A.4**, and **Appendix C** [```bbVL```, ```JZ8g```].

**2. Updated Table 1 with five runs of different random seeds:** We update the mean value in **Table 1** and the 95% confidence intervals in **Table 8** [```bbVL```].

**3. ILR with three LLMs:** We perform ILR-training on a tri-model group: Llama-3.1-8B-Instruct, Qwen2.5-7B-Instruct, and Qwen2.5-14B-Instruct. We observe a positive scaling trend of ILR and incorporate the results in **Table 5** and **Section 4.2.4** [```bbVL```, ```opn6```, ```JZ8g```].

**4. Multi-agent inference based on ILR-trained LLMs:** We empirically demonstrate that our ILR training does not diminish an agent's ability to collaborate effectively at deployment and add corresponding results in **Table 4** and **Section 4.2.2** [```opn6```].

**5. Robustness of ILR:** We discuss the heterogeneous role assignment and the scenario of Dynamic Interaction in **Appendix D** [```bbVL```, ```opn6```].

**6. More ablation Studies:** We conduct more ablation studies as suggested, including z-score normalization, periodic updates, mixed types interaction, cost analysis, comparison with more self-learning methods, and performance of the same LLM with different input prompts in **Appendix E** [```bbVL```, ```opn6```, ```JZ8g```].

We have incorporated these suggested modifications in the latest version, which are highlighted in **blue**. We are grateful for the opportunity to improve our work. If you are satisfied with the revisions, we kindly request your approval to consider an improved score.

Thanks to all the reviewers for their time again.

Best regards,

The Authors

---

### Author Response · Authors · 2025-12-02
**Summary of Rebuttal Discussion**

Dear Area Chairs, Senior Area Chairs, and Program Chairs,

We sincerely appreciate your management of the review process and thank the reviewers for their constructive feedback during the initial review phase. Their comments have motivated us to strengthen our paper. Below, we summarize how we have faithfully addressed the specific concerns of each reviewer through extensive new experiments and revisions.

## 1. To Reviewer bbVL:
- **Key Concerns**:

(1) Validity of performance gains (request confidence intervals)

(2) Comparison with self-reflection baselines

(3) Clarification on individual components

- **Our Resolution**:

(1) We re-evaluate our method using five different random seeds and update the paper with mean values and 95% confidence intervals, confirming that our improvements are statistically significant and robust.

(2) We conduct a comparison with Re-ReST (a representative self-reflection method). Results show that ILR consistently outperforms this baseline.

(3) We clarify the distinction between our Idea3 framework and existing debate paradigms, emphasizing the "critical thinking" aspect of our method.

## 2. To Reviewer opn6:

- **Key Concerns**:

(1) Comparison with advanced single-agent baselines (like CPPO)

(2) Fairness regarding sampling cost

(3) Scalability of agent numbers

(4) Performance in multi-agent inference settings

- **Our Resolution**:

(1) & (2) We compare ILR against CPPO (an advanced GRPO variant) and a 2x sampling GRPO baseline. ILR outperforms both, proving that our gains stem from the interactive learning mechanism rather than just increased sampling or computation.

(3) We extend experiments to a 3-agent setting, demonstrating a positive scaling trend.

(4) We empirically verify that ILR-trained models also perform excellently when deployed in multi-agent inference settings, addressing the concern about underutilized potential.

## 3. To Reviewer JZ8g:

- **Key Concerns**:

(1) Justification of Formula 5 (reward shaping/normalization)

(2) Dynamic updates of reasoning ability ($\gamma_i$)

(3) Feasibility of mixed-type interactions

(4) Training cost

(5) Request for pseudocode

- **Our Resolution**:

(1) We provide a detailed ablation study comparing our min-max normalization against z-score normalization (showing ours is superior) and clarified the motivation for clipping.

(2) & (3) We conduct experiments with periodic updates of $\gamma_i$ and mixed-type interactions, showing some interesting findings.

(4) We add a cost-effectiveness analysis, showing only a marginal increase in training time for significant performance gains.

(5) We add a complete pseudocode.


## Regarding Reviewer Engagement:
We posted our detailed point-by-point responses and the revised manuscript (including all new experimental results) on November 24. To ensure our responses were received, we posted two polite follow-up comments on November 27 and 28.

Regrettably, we have not received any response or acknowledgement from any of the reviewers (``bbVL``, ``opn6``, ``JZ8g``). We are disheartened by this lack of engagement, as we have made every effort to resolve the raised concerns with solid empirical evidence. We genuinely hope that our extensive additional work is not overlooked due to the silence from the reviewers.

Thank you for your support in ensuring a fair evaluation process.

Best regards,

The Authors

---

### Note · Authors · 2026-01-04

**Comment:**

We sincerely thank the Reviewers, Area Chairs, Senior Area Chairs, and Program Chairs for their time and thoughtful review. After careful consideration, we have decided to withdraw our submission.

While we have carefully addressed all reviewer concerns in our rebuttal, we regret that there has been no further communication or updated reviews, which leaves us unable to fully assess whether our revisions satisfactorily resolved the remaining issues. We believe the scholarly process benefits from constructive dialogue, and without additional feedback, we feel it is most appropriate to withdraw the paper at this stage.

**Withdrawal Confirmation:**

I have read and agree with the venue's withdrawal policy on behalf of myself and my co-authors.